# SWELL1 regulates skeletal muscle cell size, intracellular signaling, adiposity and glucose metabolism

Ashutosh Kumar[1], Litao Xie[1], Chau My Ta[1], Antentor O Hinton[2,3], Susheel K Gunasekar[1], Rachel A Minerath[2,4], Karen Shen[5], Joshua M Maurer[1], Chad E Grueter[2,4], E Dale Abel[2,3], Gretchen Meyer[5], Rajan Sah[1]*

[1]Department of Internal Medicine, Cardiovascular Division, Washington University School of Medicine, St. Louis, United States; [2]Fraternal Order of Eagles Diabetes Research Center, Iowa City, United States; [3]Division of Endocrinology and Metabolism, Iowa City, United States; [4]Division of Cardiology, University of Iowa, Iowa City, United States; [5]Program in Physical Therapy and Departments of Neurology, Biomedical Engineering and Orthopedic Surgery, Washington University in St. Louis, St. Louis, United States

**Abstract** Maintenance of skeletal muscle is beneficial in obesity and Type 2 diabetes. Mechanical stimulation can regulate skeletal muscle differentiation, growth and metabolism; however, the molecular mechanosensor remains unknown. Here, we show that SWELL1 (*Lrrc8a*) functionally encodes a swell-activated anion channel that regulates PI3K-AKT, ERK1/2, mTOR signaling, muscle differentiation, myoblast fusion, cellular oxygen consumption, and glycolysis in skeletal muscle cells. LRRC8A over-expression in *Lrrc8a* KO myotubes boosts PI3K-AKT-mTOR signaling to supra-normal levels and fully rescues myotube formation. Skeletal muscle-targeted *Lrrc8a* KO mice have smaller myofibers, generate less force ex vivo, and exhibit reduced exercise endurance, associated with increased adiposity under basal conditions, and glucose intolerance and insulin resistance when raised on a high-fat diet, compared to wild-type (WT) mice. These results reveal that the LRRC8 complex regulates insulin-PI3K-AKT-mTOR signaling in skeletal muscle to influence skeletal muscle differentiation in vitro and skeletal myofiber size, muscle function, adiposity and systemic metabolism in vivo.

*For correspondence:
rajan.sah@wustl.edu

Competing interests: The authors declare that no competing interests exist.

## Introduction

Maintenance of skeletal muscle mass and function is associated with improved metabolic health and is thought to protect against obesity and obesity-related diseases such as diabetes, nonalcoholic fatty liver disease, heart disease and osteoarthritis, and correlates with overall health in the aging population. Skeletal muscle atrophy, the loss of skeletal muscle mass, is associated with cancer (cachexia), heart failure, chronic corticosteroid use, paralysis or denervation (disuse atrophy), chronic positive pressure ventilation (diaphragm atrophy, inability to extubate), prolonged space flight (*Fitts et al., 2001*) or bed rest (unloading) and aging (*Schiaffino et al., 2013*). Each of these clinical scenarios also contribute to poor metabolic health and increase frailty. Thus, a deeper understanding of the molecular mechanisms that regulate skeletal muscle maintenance, growth and function has important implications for human health.

Skeletal muscle growth is regulated by a multitude of stimuli, including growth factor signaling, cytokines, mechanical load, integrin signaling and hormones (*Schiaffino et al., 2013*). Activation of AKT-mTOR signaling downstream of insulin and IGF-1 is well established as a critical regulator of skeletal muscle differentiation and growth (*Schiaffino et al., 2013*). However, mechanical loading of

**eLife digest** Skeletal muscles – the force-generating tissue attached to bones – must maintain their mass and health for the body to work properly. It is therefore necessary to understand how an organism can regulate the way skeletal muscles form, grow and heal.

A multitude of factors can control how muscles form, including mechanical signals. The molecules that can sense these mechanical stimuli, however, remain unknown. Mechanoresponsive ion channels provide possible candidates for these molecular sensors. These proteins are studded through the cell membranes, where they can respond to mechanical changes by opening and allowing the flow of ions in and out of a cell, or by changing interactions with other proteins.

The SWELL1 protein is a component of an ion channel known as VRAC, which potentially responds to mechanical stimuli. This channel is associated with many biological processes such as cells multiplying, migrating, growing and dying, but it is still unclear how.

Here, Kumar et al. first tested whether SWELL1 controls how skeletal muscle precursors mature into their differentiated and functional form. These experiments showed that SWELL1 regulates this differentiation process under the influence of the hormone insulin, as well as mechanical signals such as cell stretching. In addition, this work revealed that SWELL1 relies on an adaptor molecule called GRB2 to relay these signals in the cell.

Next, Kumar et al. genetically engineered mice lacking SWELL1 only in skeletal muscle. These animals had smaller muscle cells, as well as muscles that were weaker and less enduring. When raised on a high-calorie diet, the mutant mice also got more obese and developed resistance to insulin, which is an important step driving obesity-induced diabetes. Together, these findings show that SWELL1 helps to regulate the formation and function of muscle cells, and highlights how an ion channel participates in these processes.

Healthy muscles are key for overall wellbeing, as they also protect against obesity and obesity-related conditions such as type 2 diabetes or nonalcoholic fatty liver disease. This suggests that targeting SWELL1 could prove advantageous in a clinical setting.

muscle, as occurs with regular activity, exercise, and resistance training also induces mTOR-mediated skeletal muscle growth (*Ben-Sahra and Manning, 2017*; *Hornberger et al., 2006*; *Yoon, 2017*), possibly via a growth-factor independent mechanosensory mechanism. β1-integrin and focal adhesion kinase (FAK) signaling has been proposed as this mechanosensory system - sensing mechanical load on skeletal muscle and regulating hypertrophic signaling (*Carson and Wei, 2000*; *Schlaepfer et al., 1999*; *Flück et al., 1999*; *Klossner et al., 2009*). However, mechanosensitive ion channels, or ion channel complexes, may also regulate intracellular signaling, including PIEZO1, TRPV4 and the recently identified LRRC8 channel complex that functionally encodes the volume-regulated anion current (VRAC) (*Syeda et al., 2016*; *Voss et al., 2014*; *Qiu et al., 2014*).

*Lrrc8a* (leucine-rich repeat containing protein 8A, also known as SWELL1) encodes a ~ 95 kDa membrane protein with four transmembrane domains and an intracellular, C-terminal leucine-rich repeat domain (LRRD) (*Osei-Owusu et al., 2018*). It was first described, in 2003, as the site of a translocation mutation in a young woman with an immunodeficiency characterized by agammaglobulinemia and absent B-cells (*Sawada et al., 2003*; *Kubota et al., 2004*). This phenotype led to studies linking LRRC8A to lymphocyte differentiation defects arising from impaired LRRC8A dependent GRB2-mediated PI3K-AKT signaling, in part based on data from a global *Lrrc8a* knock-out (KO) mouse (*Kumar et al., 2014*). Although LRRC8A was speculated in 2012 to form a hetero-hexameric ion channel complex with other LRRC8 family members (*Abascal and Zardoya, 2012*), it was not until 2014 that LRRC8A was experimentally identified as an essential component of the volume-regulated anion channel (VRAC) (*Voss et al., 2014*; *Qiu et al., 2014*), forming hetero-hexamers with LRRC8B-E (*Syeda et al., 2016*; *Voss et al., 2014*). So, for about a decade, LRRC8A was considered a membrane protein that regulates PI3K-AKT-mediated lymphocyte function (*Sawada et al., 2003*; *Kubota et al., 2004*), putatively via non-ion channel, protein-protein interaction mediated signaling, and only later discovered to also form the long-studied VRAC ion channel signaling complex. Indeed, since its first description >30 years ago (*Cahalan and Lewis, 1988*; *Hazama and Okada, 1988*; *Pedersen et al., 2015*), VRAC has been associated with a multitude of complex physiological

and pathophysiological functions, including cell proliferation, cell migration, angiogenesis, cell death and apoptosis (*Pedersen et al., 2016*; *Eggermont et al., 2001*); however, the molecular mechanisms underlying these diverse functions had remained elusive without knowledge of the molecular identity of this ion channel complex.

We recently identified LRRC8A as a swell or stretch-activated volume sensor in adipocytes that regulates glucose uptake, lipid content, and adipocyte growth via a LRRC8A-GRB2-PI3K-AKT signaling pathway – providing a putative feed-forward amplifier to enhance adipocyte growth and insulin signaling during caloric excess (*Zhang et al., 2017*; *Gunasekar et al., 2019*). Intriguingly, others have shown that mechanical stimuli applied by pulling on β1-integrins on cardiac muscle cells with magnetic beads activates a VRAC-like current, suggesting a putative connection between β1-integrin/focal adhesion kinase signaling and LRRC8A (*Browe and Baumgarten, 2003*; *Browe and Baumgarten, 2006*). Taken together, these findings suggest that LRRC8A may connect β1-integrin-mediated mechano-transduction with insulin/IGF1-PI3K-AKT-mTOR signaling, which, in skeletal muscle is anticipated to regulate skeletal muscle differentiation, function and potentially also adiposity and systemic glucose metabolism (*Brüning et al., 1998*; *Moller et al., 1996*; *Kim et al., 2000*). In this study, we tested this hypothesis by examining LRRC8A dependent intracellular signaling and myotube differentiation in both C2C12 and primary skeletal muscle cells in vitro, and by performing skeletal muscle targeted LRRC8A loss-of-function experiments in vivo. We find that myotube differentiation and insulin and stretch-mediated PI3K-AKT, ERK1/2, mTOR signaling is strongly regulated by LRRC8A protein expression, and provide evidence that GRB2 signaling mediates these LRRC8A dependent effects. Finally, using skeletal muscle *Lrrc8a* knock-out mice, we reveal the requirement of skeletal muscle LRRC8A for maintaining normal skeletal muscle cell size, muscle endurance, force generation, adiposity and glucose tolerance, under basal conditions and in the setting of overnutrition.

## Results

### LRRC8A is expressed and functional in skeletal muscle and is required for myotube formation

LRRC8A is the essential component of a hexameric ion channel signaling complex that encodes $I_{Cl,SWELL}$, or the volume-regulated anion current (VRAC) (*Voss et al., 2014*; *Qiu et al., 2014*). While the LRRC8A complex has been shown to regulate cellular volume in response to application of non-physiological hypotonic extracellular solutions, the physiological function(s) of this ubiquitously expressed ion channel signaling complex remain unknown. To determine the function of the LRRC8A channel complex in skeletal muscle, we genetically deleted *Lrrc8a* from C2C12 mouse myoblasts using CRISPR/cas9-mediated gene editing as described previously (*Zhang et al., 2017*; *Kang et al., 2018*), and from primary skeletal muscle cells isolated from *Lrrc8a* $^{fl/fl}$ mice transduced with adenoviral Cre-mCherry (KO) or mCherry alone (WT control) (*Zhang et al., 2017*). LRRC8A protein western blots confirmed robust *Lrrc8a* ablation in both *Lrrc8a* KO C2C212 myotubes and *Lrrc8a* KO primary skeletal myotubes (*Figure 1A*). Next, whole-cell patch clamp revealed that the hypotonically activated (210 mOsm) outwardly rectifying current present in WT C2C12 myoblasts is abolished in *Lrrc8a* KO C2C12 myoblasts (*Figure 1B*), confirming LRRC8A as also required for $I_{Cl,SWELL}$ or VRAC in skeletal muscle myoblasts. Remarkably, *Lrrc8a* ablation is associated with impaired myotube formation in both C2C12 myoblasts and in primary skeletal satellite cells (*Figure 1C*), with an 58% and 45% reduction in myotube area in C2C12 and skeletal muscle myotubes, respectively, compared to WT. As an alternative metric, myoblast fusion is also markedly reduced by 80% in *Lrrc8a* KO C2C12 compared to WT, as assessed by myotube fusion index (number of nuclei inside myotubes/total number of nuclei; *Figure 1C*).

### Global transcriptome analysis reveals that *Lrrc8a* ablation blocks myogenic differentiation and dysregulates multiple myogenic signaling pathways

In order to further characterize the observed LRRC8A-dependent impairment in myotube formation in C2C12 and primary muscle cells, we performed genome-wide RNA sequencing (RNA-seq) of *Lrrc8a* KO C2C12 relative to control WT C2C12 myotubes. These transcriptomic data revealed clear

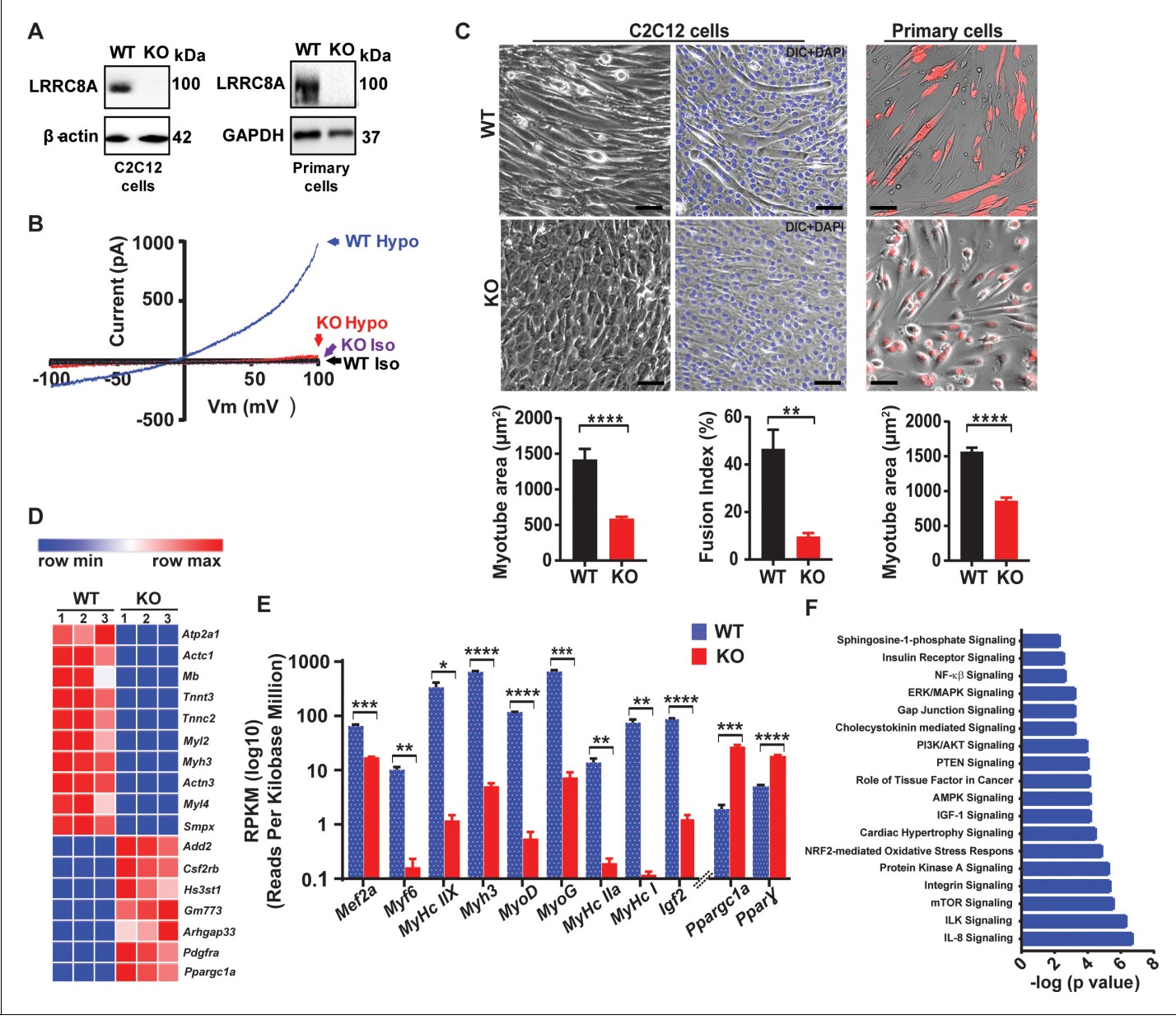

**Figure 1.** Skeletal muscle LRRC8A is required for myotube formation and regulates multiple myogenic signaling pathways. (**A**) Western blots from WT and *Lrrc8a* KO C2C12 (KO) (left) and primary myotubes (right). (**B**) Current-voltage curves from WT and *Lrrc8a* KO C2C12 myoblast measured during a voltage-ramp from −100 to +100 mV +/- isotonic and hypotonic (210 mOsm) solution. (**C**) Bright field merged with fluorescence images of differentiated WT and *Lrrc8a* KO C2C12 myotubes (left, middle) and skeletal muscle primary cells (right). DAPI stains nuclei blue (middle). Red is mCherry reporter fluorescence from adenoviral transduction. Scale bar: 100 μm. Mean myotube surface area measured from WT (n = 21) and *Lrrc8a* KO (n = 21) C2C12 myotubes (left), and WT (n = 22) and *Lrrc8a* KO (n = 15) primary skeletal myotubes (right). Fusion index (%multinucleated cells) measured from WT (n = 5 fields) and *Lrrc8a* KO (n = 5 fields) C2C12 (shown below the representative image). (**D**) Heatmap of top 17 differentially expressed genes in WT versus *Lrrc8a* KO C2C12 myotubes derived from RNA sequencing. (**E**) Reads Per Kilobase Million for select myogenic differentiation genes (n = 3, each). (**F**) IPA canonical pathway analysis of genes significantly regulated in *Lrrc8a* KO C2C12 myotubes in comparison to WT. n = 3 for each group. For analysis with IPA, FPKM cutoffs of 1.5, fold change of ≥1.5, and false discovery rate <0.05 were utilized for significantly differentially regulated genes. Statistical significance between the indicated values were calculated using a two-tailed Student's t-test. Error bars represent mean ± s.e.m. *, p<0.05, **, p<0.01, ***, p<0.001, ****, p<0.0001. n = 3, independent experiments.

differences in the global transcriptional profile between WT and *Lrrc8a* KO C2C12 myotubes (*Figure 1D* and *Supplementary file 1*), with marked suppression of numerous skeletal muscle differentiation genes including *Mef2a* (0.2-fold), *Myl2* (0.008-fold), *Myl3* (0.01-fold), *Myl4* (0.008-fold), *Actc1* (0.005-fold), *Tnnc2* (0.005-fold), *Igf2* (0.01-fold) (*Figure 1E*). Curiously, this suppression of myogenic differentiation is associated with marked induction of *Ppargc1α* (PGC1α; 14-fold) and *Pparγ* (3.7-fold). PGC1α and *Pparγ* are positive regulators of skeletal muscle differentiation (*Haralampieva et al., 2017*; *Ruas et al., 2012*; *Singh et al., 2007*), suggesting that the LRRC8A-dependent defect in skeletal muscle differentiation lies downstream of PGC1α and PPARγ. To further define putative pathway dysregulation underlying disruptions in myogenesis observed in *Lrrc8a* KO C2C12 myotubes, we next performed pathway analysis on the transcriptome data. We found that numerous signaling pathways essential for myogenic differentiation are disrupted, including insulin ($2 \times 10^{-3}$), MAP kinase ($5 \times 10^{-4}$), PI3K-AKT ($1 \times 10^{-4}$), AMPK ($6 \times 10^{-5}$), integrin ($3 \times 10^{-6}$), mTOR ($2 \times 10^{-6}$), integrin linked kinase ($4 \times 10^{-7}$) and IL-8 ($1 \times 10^{-7}$) signaling pathways (*Figure 1F* and *Supplementary file 2*).

## LRRC8A regulates multiple insulin-dependent signaling pathways in skeletal myotubes

Guided by the results of the pathway analysis, and the fact that skeletal myogenesis and maturation is regulated by insulin-PI3K-AKT-mTOR-MAPK (*Conejo et al., 2001*; *Schiaffino et al., 2013*), we directly examined a number of insulin-stimulated pathways in WT and *Lrrc8a* KO C2C12 myotubes, including insulin-stimulated AKT2-AS160, FOXO1 and AMPK signaling. Indeed, insulin-stimulated pAKT2, pAS160, pFOXO1 and pAMPK are abrogated in *Lrrc8a* KO myotubes compared to WT C2C12 myotubes (*Figure 2A and C*). Importantly, insulin-AKT-AS160 signaling is also diminished in *Lrrc8a* KO primary skeletal muscle myotubes compared to WT primary myotubes (*Figure 2B and D*), consistent with the observed differentiation block (*Figure 1C*). This confirms that LRRC8A-dependent insulin-AKT and downstream signaling is not a feature specific to immortalized C2C12 myotubes but is also conserved in primary skeletal myotubes. It is also notable that reduction in total AKT2 protein is associated with *Lrrc8a* ablation in both C2C12 and primary skeletal muscle cells, and this is consistent with threefold reduction in AKT2 mRNA expression observed in RNA sequencing data (*Figure 2E*). Moreover, transcription of a number of critical insulin signaling and glucose homeostatic genes are suppressed by *Lrrc8a* ablation, including GLUT4 (*Slc2a4*, 51-fold), FOXO3 (2-fold), FOXO4 (2.8-fold) and FOXO6 (18-fold) (*Figure 2E*). Indeed, FOXO signaling is thought to integrate insulin signaling with glucose metabolism (*Lee and Dong, 2017*; *Gross et al., 2008*) in a number of insulin sensitive tissues. Collectively, these data indicate that impaired LRRC8A-dependent insulin-AKT-AS160-FOXO signaling is associated with the observed defect in myogenic differentiation upon *Lrrc8a* ablation in cultured skeletal myotubes, and also predict putative impairments in skeletal muscle glucose metabolism and oxidative metabolism.

We next asked if these LRRC8A-dependent effects on downstream skeletal myotube insulin signaling are due to impaired myotube differentiation, and associated impairments in insulin signaling, or if LRRC8A also regulates these signaling pathways in differentiated skeletal myotubes. To test this, we first fully differentiated C2C12 myotubes, and then knocked down (KD) LRRC8A using Ad-shLRRC8A-mCherry post-differentiation and compared myotube size and insulin-stimulated signaling to Ad-shSCR-mCherry treated C2C12 myotubes (*Figure 2—figure supplement 1*). Similar to *Lrrc8a* ablation prior to differentiation, shRNA-mediated LRRC8A KD of fully differentiated C2C12 myotubes mildly reduced myotube size (*Figure 2—figure supplement 1A*), and significantly reduced insulin-stimulated pAKT2, pAKT1, pAS160, pAMPK and pS6 ribosomal proteins (*Figure 2—figure supplement 1B and C*), although in some cases, these differences were less marked than in C2C12 *Lrrc8a* KO myoblasts that were subsequently differentiated (*Figure 2*). Also, the reductions in total AKT2 observed at both the mRNA (*Figure 2E*, and *Supplementary file 1*) and protein levels (*Figure 2A*) in differentiated *Lrrc8a* KO C2C12 myoblasts are also observed upon LRRC8A KD in differentiated C2C12 myotubes (*Figure 2—figure supplement 1B and C*), suggesting that total AKT2 expression is regulated by LRRC8A in fully differentiated myotubes. As a complementary approach, we confirmed these findings in fully differentiated, primary skeletal myotubes isolated from *Lrrc8a*[fl/fl] mice upon adenoviral transduction of Ad-CMV-Cre-mCherry (KO) compared to Ad-CMV-mCherry (WT), and noted similar reductions in myotube size (*Figure 2—figure supplement 1D*), and in basal levels of pAS160, pAKT2, AKT2 and pAKT1 signaling (*Figure 2—figure supplement 1E and F*).

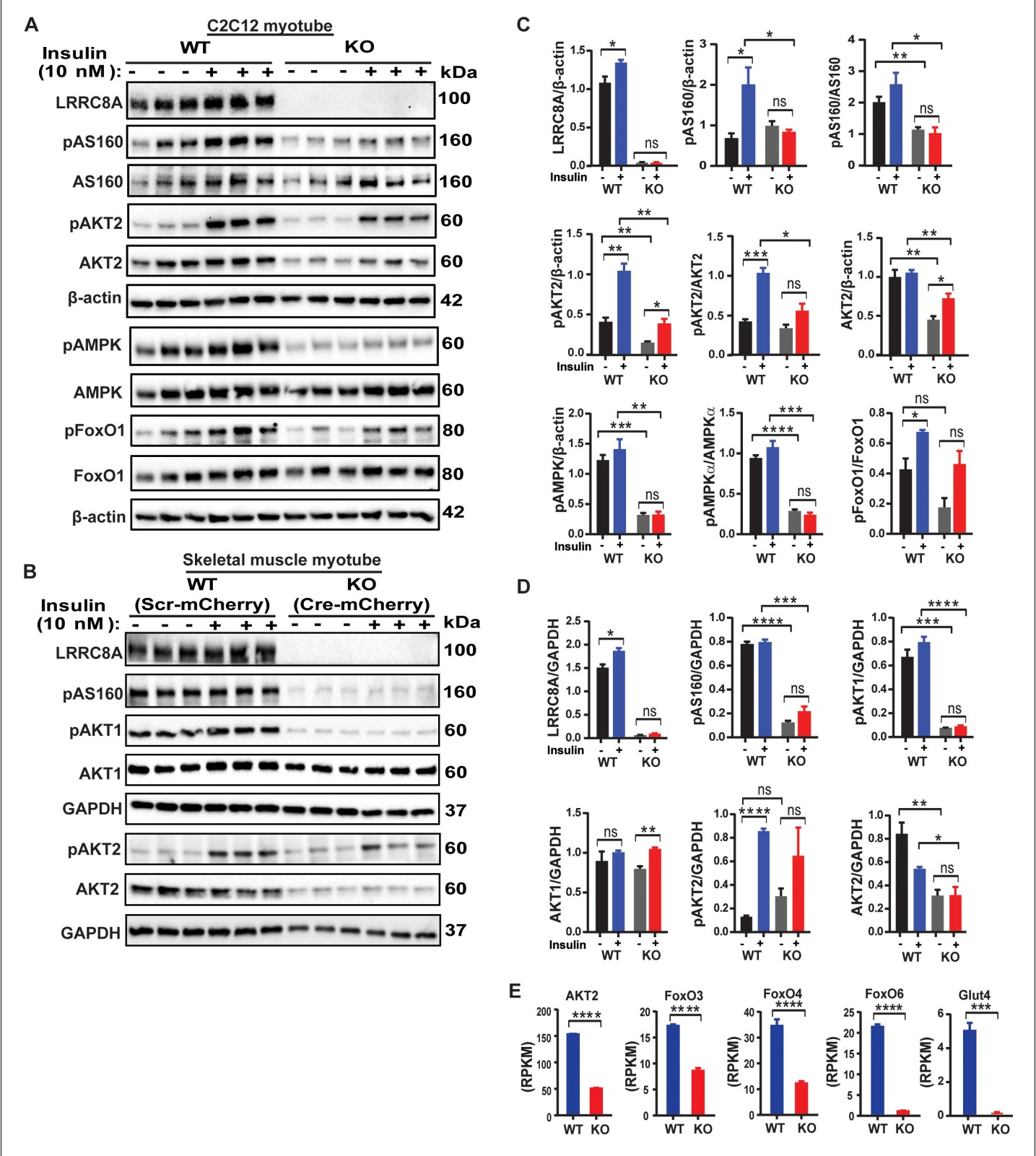

**Figure 2.** LRRC8A regulates multiple insulin dependent signaling pathways in skeletal myotubes. (**A**) Western blots of LRRC8A, pAKT2, AKT2, pAS160, AS160, pAMPK, AMPK, pFoxO1, FoxO1 and β-actin in WT and *Lrrc8a* KO C2C12 myotubes upon insulin-stimulation (10 nM). (**B**) Western blots of LRRC8A, pAS160, pAKT1, AKT1, pAKT2, AKT2 and GAPDH in WT (Ad-CMV-mCherry) and *Lrrc8a* KO (Ad-CMV-Cre-mCherry) primary skeletal muscle myotubes following insulin-stimulation (10 nM). (**C** and **D**) Densitometric quantification of proteins depicted on western blots normalized to

*Figure 2 continued on next page*

Figure 2 continued

corresponding β-actin and GAPDH respectively. (E) Gene expression analysis of insulin signaling associated genes AKT2, FOXO3, FOXO4, FOXO6 and GLUT4 in WT and *Lrrc8a* KO C2C12 myotubes. Statistical significance between the indicated values were calculated using a two-tailed Student's t-test. Error bars represent mean ± s.e.m. *, p<0.05, **, p<0.01, ***, p<0.001, ****, p<0.0001. n = 3, independent experiments.

The online version of this article includes the following figure supplement(s) for figure 2:

**Figure supplement 1.** LRRC8A regulates myotube size and insulin dependent signaling pathways in differentiated skeletal myotubes.

Taken together, these data indicate that impaired insulin signaling observed in LRRC8A-depleted myotubes is in part due to impaired myotube differentiation with secondary effects on insulin signaling, and in part due to a direct contribution of LRRC8A to intracellular signaling in fully differentiated myotubes.

## LRRC8A over-expression in LRRC8A depleted C2C12 is sufficient to rescue myogenic differentiation and augment intracellular signaling above baseline levels

To further validate LRRC8A-mediated effects on muscle differentiation and signaling, we re-expressed LRRC8A in *Lrrc8a* KO C2C12 myoblasts ( LRRC8A O/E) and then examined myotube differentiation and basal activity of multiple intracellular signaling pathways by western blot, including pAKT1, pAKT2, pAS160, p-p70S6K, pS6K and pERK1/2 as compared to WT and *Lrrc8a* KO C2C12 myotubes. LRRC8A O/E to 2.12-fold WT levels fully rescues myotube development in *Lrrc8a* KO myotubes (*Figure 3A*), as quantified by restoration of *Lrrc8a* KO myotube area to levels above WT (*Figure 3B*). This rescue of *Lrrc8a* KO myotube development upon LRRC8A O/E (*Figure 3A&B*) is associated with either restored (pAS160, AKT2, pAKT1, AKT1, p70S6K) or supra-normal (pAKT2, p-p70S6K, pS6K, pERK1/2) signaling (*Figure 3C&D*) compared to WT C2C12 myotubes. These data demonstrate that LRRC8A protein expression level strongly regulates skeletal muscle insulin signaling and myogenic differentiation.

## LRRC8A mediates stretch-dependent PI3K-pAKT2-pAS160-MAPK signaling in C2C12 myotubes

In a cellular context, there are numerous reports that VRAC and the LRRC8A complex that functionally encodes it is mechano-responsive (*Browe and Baumgarten, 2003*; *Browe and Baumgarten, 2006*; *Osei-Owusu et al., 2018*; *Barakat et al., 1999*; *Nakao et al., 1999*; *Nilius and Droogmans, 2001*; *Romanenko et al., 2002*; *Strange et al., 2019*). It is well established that mechanical stretch is an important regulator of skeletal muscle proliferation, differentiation and skeletal muscle hypertrophy and may be mediated by PI3K-AKT-MAPK signaling (*Schiaffino et al., 2013*; *Ma et al., 2017*; *Fu et al., 2018*) and integrin signaling pathways (*Carson and Wei, 2000*; *Schlaepfer et al., 1999*; *Flück et al., 1999*; *Klossner et al., 2009*). To determine if LRRC8A is also required for stretch-mediated AKT and MAP kinase signaling in skeletal myotubes, we subjected WT and *Lrrc8a* KO C2C12 myotubes to 0% or 5% equiaxial stretch using the FlexCell stretch system. Mechanical stretch (5%) is sufficient to stimulate PI3K-AKT2/AKT1-pAS160-MAPK (ERK1/2) signaling in WT C2C12 in a LRRC8A-dependent manner (*Figure 4A and B*). These data position LRRC8A as a co-regulator of both insulin and stretch-mediated PI3K-AKT- pAS160-MAPK signaling.

## LRRC8A interacts with GRB2 in C2C12 myotubes and regulates myogenic differentiation

It has been reported earlier in both lymphocyte and adipocytes that the LRRC8A complex interacts with Growth factor Receptor-Bound 2 (GRB2) and regulates PI3K-AKT signaling (*Kumar et al., 2014*; *Zhang et al., 2017*; *Gunasekar et al., 2019*), whereby GRB2 binds with IRS1/2 and negatively regulates insulin signaling (*Shan et al., 2013*). Indeed, GRB2 knock-down augments insulin-PI3K-MAPK signaling and induces myogenesis and myogenic differentiation genes (*Shan et al., 2013*; *Liu et al., 2009*; *Mitra and Thanabalu, 2017*). To determine if LRRC8A and GRB2 interact in C2C12 myotubes, we overexpressed C-terminal 3XFlag tagged LRRC8A in C2C12 cells followed by immunoprecipitation (IP) with Flag antibody. We observed significant GRB2 enrichment upon Flag IP from lysates of LRRC8A-3xFlag expressing C2C12 myotubes, consistent with a GRB2-LRRC8A interaction

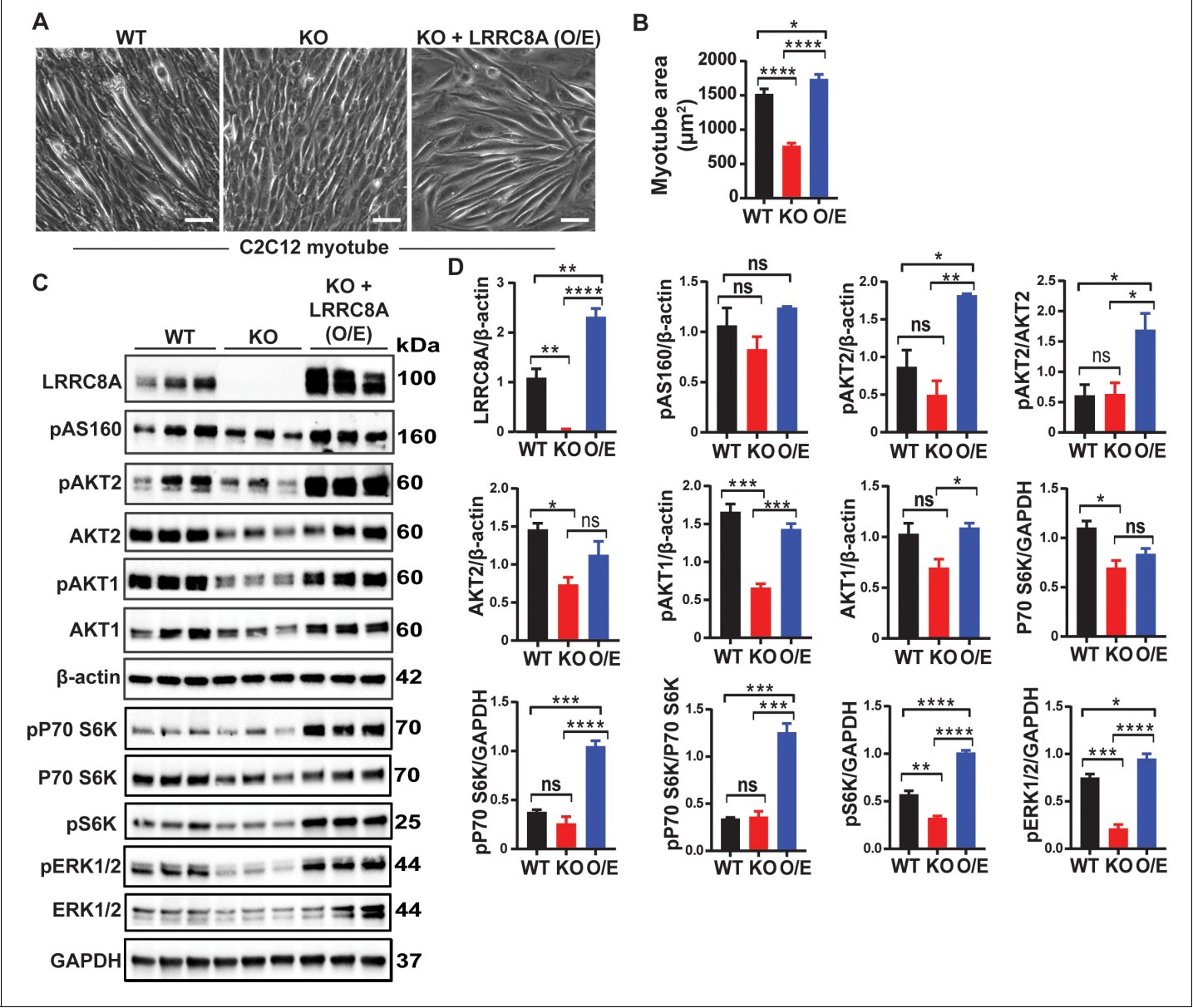

**Figure 3.** LRRC8A over-expression in *Lrrc8a* KO C2C12 myotubes rescues myogenic differentiation and augments intracellular signaling. (**A**) Bright-field image of differentiated WT, *Lrrc8a* KO (KO) and *Lrrc8a* KO + LRRC8A O/E C2C12 myotubes. (**A**) Quantification of mean myotube surface areas in WT (n = 35), *Lrrc8a* KO C2C12 (n = 26) and *Lrrc8a* KO + LRRC8A O/E C2C12 (n = 45) cells. Scale bar: 100 µm. (**C**) Western blots of LRRC8A, AKT2, pAKT2, pAS160, pAKT1, AKT1, pP70S6K, P70S6K, pS6K, pERK1/2, ERK1/2, β-actin and GAPDH from WT, *Lrrc8a* KO and *Lrrc8a* KO + LRRC8A O/E C2C12 myotubes. (**D**) Densitometric quantification of proteins depicted on western blots normalized to β-actin and GAPDH, respectively. Statistical significance between the indicated group were calculated with one-way ANOVA, Tukey's multiple comparisons test. Error bars represent mean ± s.e.m. *, p<0.05, **, p<0.01, ***, p<0.001, ****, p<0.0001. n = 3, independent experiments.

(*Figure 5A*). Based on the notion that LRRC8A titrates GRB2-mediated suppression of AKT/MAPK signaling, and that *Lrrc8a* ablation results in unrestrained GRB2-mediated AKT/MAPK inhibition (*Gunasekar et al., 2019*), we next tested if GRB2 knock-down (KD) may rescue myogenic differentiation in *Lrrc8a* KO C2C12 myotubes. shRNA-mediated GRB2 KD in *Lrrc8a* KO C2C12 myoblasts ( LRRC8A KO/shGRB2; *Figure 5B*) stimulates myotube formation (*Figure 5C*) and increases myotube area (*Figure 5D*), to levels equivalent to WT/shSCR (*Figure 5C and D*). Similarly, GRB2 KD in *Lrrc8a* KO C2C12 myotubes induces myogenic differentiation markers IGF1, MyoHCl, MyoHCIIa and MyoH-CIIb relative to both LRRC8A KO/shSCR and WT/shSCR (*Figure 5E and F*). These data are consistent

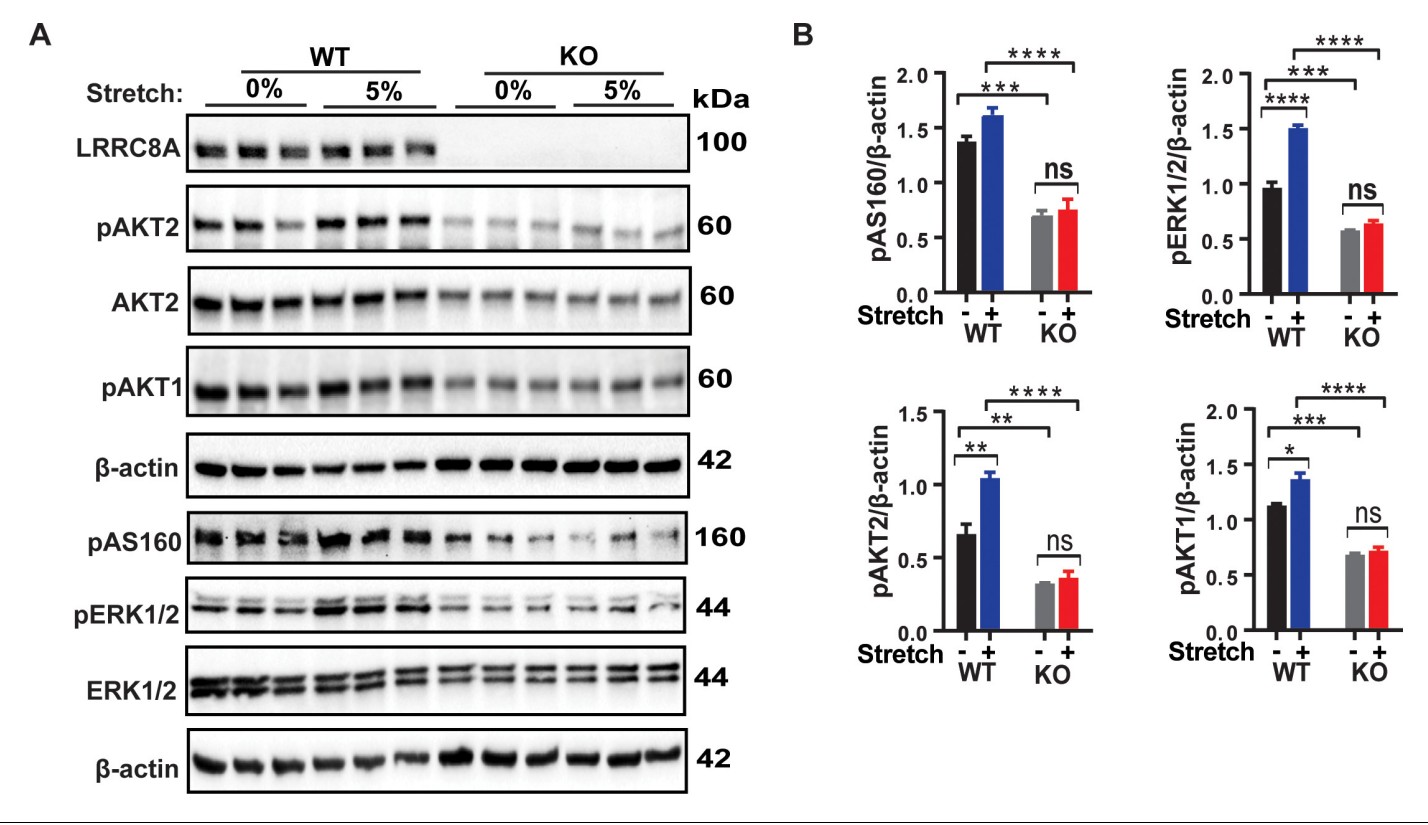

**Figure 4.** LRRC8A is required for intact stretch-induced PI3K-pAKT2-pAS160-MAPK signaling in C2C12 myotubes. (**A**) Western blot of LRRC8A, AKT2, pAKT2, pAKT1, pAS160, pERK1/2, ERK1/2 and β-actin in WT and *Lrrc8a* KO myotube in response to 15 min of 0% and 5% static stretch. (**B**) Densitometric quantification of each signaling protein relative to β-actin. Statistical significance between the indicated group calculated with one-way Anova, Tukey's multiple comparisons test. Error bars represent mean ± s.e.m. *, p<0.05, **, p<0.01, ***, p<0.001, ****, p<0.0001. n = 3, independent experiments.

with GRB2 suppression rescuing myotube differentiation in *Lrrc8a* KO C2C12, and supports a model in which LRRC8A regulates myogenic differentiation by titrating GRB2-mediated signaling.

### Skeletal muscle-targeted *Lrrc8a* knock-out mice have reduced skeletal myocyte size, muscle endurance and ex vivo force generation

To examine the physiological consequences of *Lrrc8a* ablation in vivo, we generated skeletal muscle-specific *Lrrc8a* KO mice using Cre-LoxP technology by crossing *Myf5-Cre* mice with *Lrrc8a^{fl/fl}* mice (Skm KO; *Figure 6A*), and confirmed robust skeletal muscle LRRC8A depletion in Skm KO gastrocnemius muscle, 12.3-fold lower than WT controls (*Figure 6B*). Remarkably, in contrast to the severe impairments in skeletal myogenesis observed in both *Lrrc8a* KO C2C12 and primary skeletal myotubes in vitro (*Figures 1*, *3* and *5*), Skm KO mice develop skeletal muscle mass comparable to WT littermates, based on Echo/MRI body composition (*Figure 6C*) and gross muscle weights (*Figure 6D*), and are born at normal mendelian ratios (*Supplementary file 3*). However, histological examination reveals a 27% reduction in skeletal myocyte cross-sectional area in Skm KO as compared to WT mice (*Figure 6E*), suggesting a requirement for *LRRC8A* in skeletal muscle cell size regulation in vivo. This is potentially due to reductions in myotube fusion, as observed in C2C12 and primary skeletal muscle cells in vitro (*Figure 1*), but occurring to a lesser degree in vivo. These data indicate that the profound impairments in myogenesis observed in vitro may reflect a very early requirement for LRRC8A signaling in skeletal muscle development (prior to LRRC8A protein elimination by *Myf5-Cre*-mediated *LRRC8A* recombination), or other fundamental differences in myogenic differentiation processes in vitro versus in vivo.

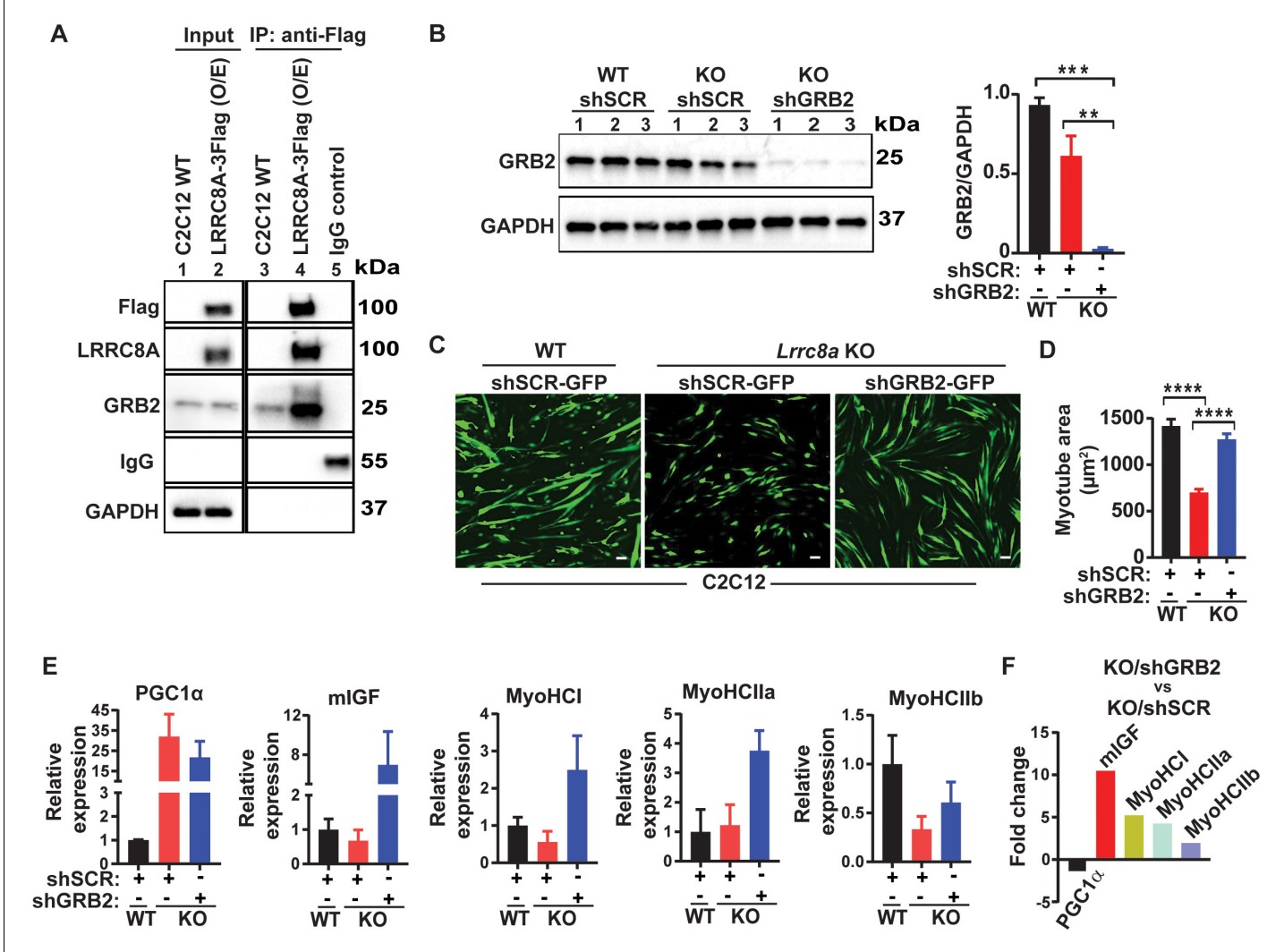

**Figure 5.** LRRC8A interacts with GRB2 in C2C12 myotubes and regulates myogenic differentiation. (**A**) LRRC8A-3xFlag over expressed in C2C12 cells followed by immunoprecipitation (IP) with Flag antibody. Western blot of Flag, LRRC8A, GRB2 and GAPDH. IgG serves as a negative control. (**B**) Western blot of GRB2 to validate GRB2 knock down efficiency in *Lrrc8a* KO/GRB2 knock-down (Ad-shGRB2-GFP) compared to WT C2C12 (Ad-shSCR-GFP) and *Lrrc8a* KO (Ad-shSCR-GFP). Densitometric quantification of GRB2 knock-down relative to GAPDH (right). (**C**) Fluorescence image of WT C2C12/shSCR-GFP, *Lrrc8a* KO/shSCR-GFP and *Lrrc8a* KO/shGRB2-GFP myotubes. Scale bar: 100 μm. (**D**) Quantification of mean myotube area of WT C2C12/shSCR-GFP (n = 25), *Lrrc8a* KO/shSCR-GFP (n = 28) and *Lrrc8a* KO/shGRB2-GFP (n = 24). (**E**) Relative mRNA expression of selected myogenic differentiation genes in *Lrrc8a* KO/shSCR and *Lrrc8a* KO/shGRB2 compared to WT C2C12/shSCR (n = 3 each), and of *Lrrc8a* KO/shGRB2 compared to *Lrrc8a* KO/shSCR (**F**), fold change of mRNA's in KO shGRB2 relative to KO cells with preserved GRB2 expression. Statistical significance between the indicated group were calculated with one-way ANOVA, Tukey's multiple comparisons test. Error bars represent mean ± s.e.m. *, p<0.05, **, p<0.01, ***, p<0.001, ****, p<0.0001. n = 3, independent experiments.

Since insulin signaling is an important regulator of skeletal muscle oxidative capacity and endurance (*Affourtit, 2016*), we next examined exercise tolerance on treadmill testing in *Lrrc8a*$^{fl/fl}$ (WT) compared to Skm KO . Skm KO mice exhibit a 14% reduced exercise capacity, compared to age and gender matched WT controls (*Figure 7A*; *Figure 7—Video 1*). Hang-times on inversion testing are also reduced 29% in Skm KO compared to controls, further supporting reduced skeletal muscle endurance upon skeletal muscle LRRC8A depletion in vivo (*Figure 7B*). To determine if these reductions in muscle function in vivo are due to muscle-specific functional impairments, we next performed ex vivo experiments in which we isolated the soleus muscle from mice and performed twitch/train testing. We observed that peak developed tetanic tension is 15% reduced in Skm KO soleus muscle compared to WT controls (*Figure 7C*), suggesting a skeletal muscle autonomous

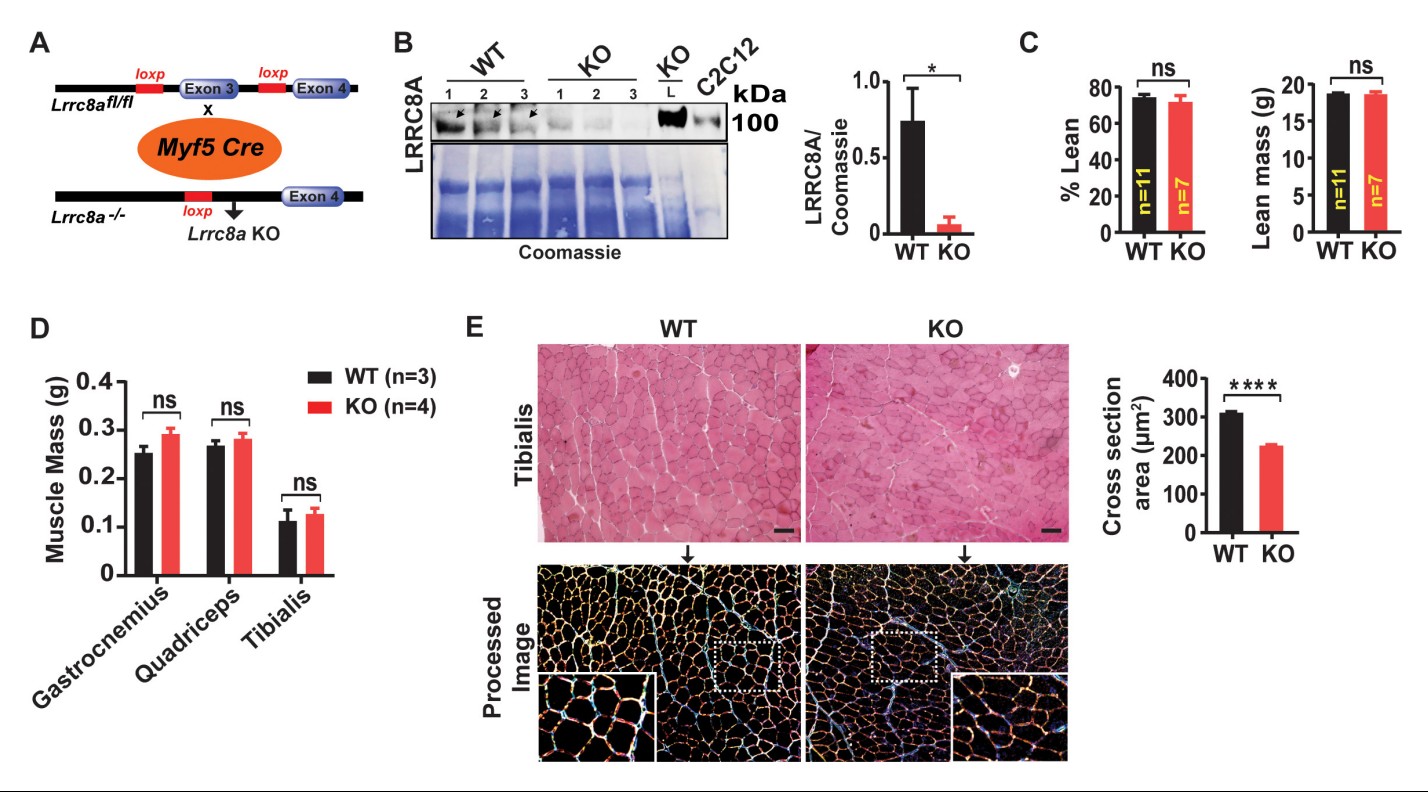

**Figure 6.** Skeletal muscle-targeted *Lrrc8a* KO mice develop smaller myofibers but normal muscle mass. (**A**) Schematic representation of Cre-mediated recombination of loxP sites flanking Exon three using muscle-specific *Myf5-Cre* mice to generate skeletal muscle targeted *Lrrc8a* KO mice. (**B**) Western blot of gastrocnemius muscle protein isolated from of WT and *Myf5$^{Cre}$/Lrrc8a$^{fl/fl}$* (*KO*) mice. Liver sample from *KO* and C2C12 cell lysates used as a positive control for LRRC8A. Coomassie gel, below, serves as loading control for skeletal muscle protein. Densitometric quantification for LRRC8A deletion in skeletal muscle of *KO* mice (n = 3) compared to WT (n = 3; *Lrrc8a$^{fl/fl}$*) (right). (**C**) NMR measurement of lean mass (%) and absolute fat mass of WT (n = 11) and *KO* (n = 7) mice. (**D**) Absolute muscle mass of muscle groups freshly isolated from WT (n = 3) and *KO* (n = 4). (**E**) Hematoxylin and eosin staining of tibialis muscle of WT and *KO* mice fed on regular chow diet for 28 weeks (above). Scale bar: 100 μm. Below, ImageJ converted image highlights distinct surface boundaries of myotubes. Inset, enlarged image shows smaller fiber size in *KO* muscle tissue. Quantification of average cross-sectional area of muscle fiber of WT (n = 300) and *KO* (n = 300) mice from 10 to 12 different view field images (right). Statistical significance between the indicated values were calculated using a two-tailed Student's t-test. Error bars represent mean ± s.e.m. *, p<0.05, **, p<0.01, ***, p<0.001, ****, p<0.0001.

mechanism, with no difference in time to fatigability (TTF, *Figure 7D*) or time to 50% decay (*Figure 7E*).

To determine whether these LRRC8A dependent differences in muscle endurance and force were due to impaired oxidative capacity, we next measured oxygen consumption rate (OCR) and extracellular acidification rate (ECAR) in WT and *Lrrc8a* KO primary skeletal muscle cells, under basal and insulin-stimulated conditions (*Figure 7F*). Oxygen consumption of *Lrrc8a* KO primary myotubes are 26% lower than WT and, in contrast to WT cells, are unresponsive to insulin-stimulation (*Figure 7F*), consistent with abrogation of insulin-AKT/ERK1/2 signaling upon skeletal muscle LRRC8A depletion. These relative changes persist in the presence of Complex V and III inhibitors, Oligomycin and Antimycin A (*Figure 7F and G*), suggesting that insulin-stimulated glycolytic pathways are primarily dysregulated upon LRRC8A depletion. In contrast, FCCP, which maximally uncouples mitochondria, abolishes differences in oxygen consumption between WT and *Lrrc8a* KO primary muscle cells, suggesting no differences in functional mitochondrial content in *Lrrc8a* KO muscle. Consistent with this finding, we observe no differences in muscle fiber-type based on Myosin Heavy Chain (MHC) Type 1, MHC Type IIa, MHC Type IIx and MHC Type IIb in both superficial and deep tibialis anterior muscle in Skm KO as compared to WT mice (*Figure 7—figure supplement 1*), indicating the reductions in muscle force and endurance in Skm KO mice is not due to muscle fiber-type switching. Indeed, it is increasingly appreciated that changes in muscle metabolic potential and morphology may occur

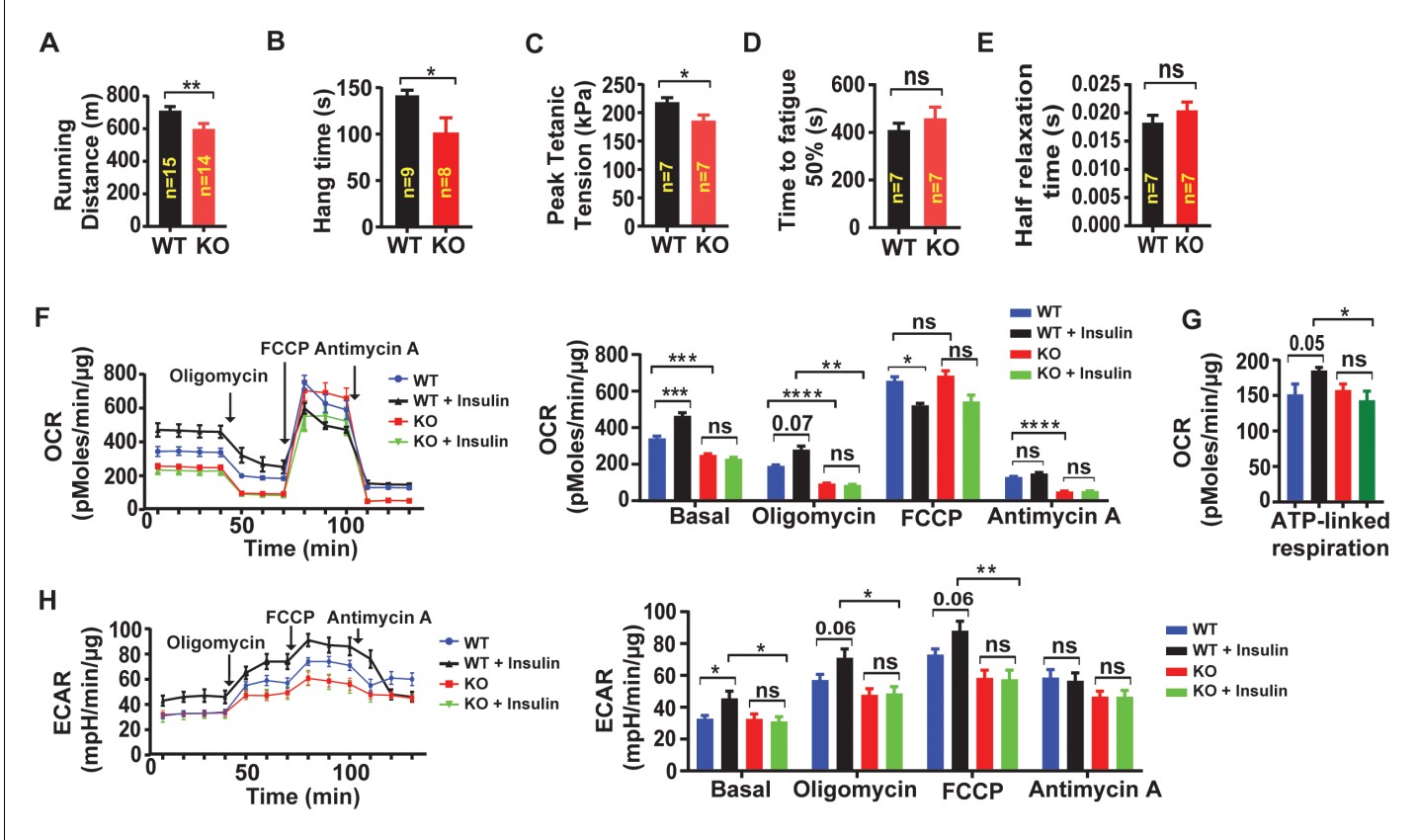

**Figure 7.** Skeletal muscle-targeted LRRC8A deletion impairs muscle endurance, force generation and insulin-stimulated oxygen consumption. (**A**) Exercise treadmill tolerance test for *KO* mice (n = 14) compared to WT littermates (n = 15). (**B**) Hang times on inversion testing of *KO* (n = 8) and WT (n = 9) mice. (**C–E**) Ex vivo isometric peak tetanic tension (**C**), time to fatigue (**D**) and half relaxation time (**E**) of isolated soleus muscle from *KO* (n = 7) compared to WT (n = 7) mice. (**F**) Oxygen consumption rate (OCR) in WT and *Lrrc8a* KO primary myotubes +/- insulin stimulation (10 nM) (n = 6 independent experiments) and quantification of basal OCR, OCR post-Oligomycin, OCR post-FCCP and OCR post-Antimycin A. (**G**) ATP-linked respiration obtained by subtracting the OCR after oligomycin from baseline cellular OCR. (**H**) Extracellular acidification rate (ECAR) in WT and *Lrrc8a* KO primary myotubes +/- insulin stimulation (10 nM) (n = 6 independent experiments) and quantification of basal OCR, OCR-post Oligomycin, OCR-post FCCP and OCR post-Antimycin A. Statistical significance between the indicated values were calculated using a two-tailed Student's t-test. Error bars represent mean ± s.e.m. *, p<0.05, **, p<0.01, ***, p<0.001, ****, p<0.0001.

The online version of this article includes the following video and figure supplement(s) for figure 7:

**Figure supplement 1.** Quantification of skeletal muscle fiber type in WT and Skm KO TA muscle.

**Figure supplement 2.** *Lrrc8a* ablation in C2C12 myotubes downregulates glucose and glycogen-associated genes.

**Figure 7—video 1.** Exercise Treadmill Testing of WT and *Myf5-Cre/ SWELL1flfl* mice.

https://elifesciences.org/articles/58941#fig7video1

independent from adaptive fiber-type switching as measured by a change in MHC expression (***Egan and Zierath, 2013***).

To examine an alternative mechanism, such as glycolysis, we measured extracellular acidification rate (ECAR) in WT and *Lrrc8a* KO primary myotubes. Insulin-stimulated ECAR increases are abolished in *Lrrc8a* KO compared to WT cells, and these differences persist independent of electron transport chain modulators (***Figure 7H***). These data suggest that LRRC8A regulation of skeletal muscle cellular oxygen consumption occurs at the level of glucose metabolism - potentially via LRRC8A-dependent insulin-PI3K-AKT-AS160-GLUT4 signaling, glucose uptake and utilization. These findings in primary skeletal muscle cells are supported by marked transcriptional suppression of numerous glycolytic genes: *Aldoa*, *Eno3*, *Pfkm*, and *Pgam2*; and glucose and glycogen metabolism genes: *Phka1*, *Phka2*, *Ppp1r3c* and *Gys1*, upon *Lrrc8a* ablation in C2C12 myotubes (***Figure 7—figure supplement 2***).

# Skeletal muscle targeted *Lrrc8a* ablation impairs systemic glucose metabolism and increases adiposity

Guided by evidence of impaired insulin-PI3K-AKT-AS160-GLUT4 signaling observed in *Lrrc8a* KO C2C12 and primary myotubes, we next examined systemic glucose homeostasis and insulin sensitivity in WT and Skm KO mice by measuring glucose and insulin tolerance. On a regular chow diet, there are no differences in either glucose tolerance or insulin tolerance (*Figure 8A*) between WT and Skm KO mice. However, over the course of 16–24 weeks on chow diet Skm KO mice develop 29% increased adiposity based on body composition measurements (*Figure 8B*) compared to WT, with no significant difference in lean mass (*Figure 8C*) or in total body mass (*Figure 8C*). When Skm KO mice are raised on a high-fat-diet (HFD) for 16 weeks, there is no difference in adiposity observed (*Figure 8—figure supplement 1*) compared to WT mice, but glucose tolerance is impaired (*Figure 8D*) and there is mild insulin resistance in HFD Skm KO mice as compared to WT (*Figure 8E*).

Since *Myf5* is also expressed in brown fat (*Seale et al., 2008*), it is possible that these metabolic phenotypes arise from LRRC8A-mediated effects in brown fat and consequent changes in systemic metabolism. To rule out this possibility, we repeated a subset of the above experiments in a skeletal muscle-targeted KO mouse generated by crossing the *Myl1-Cre* and *Lrrc8a*<sup>fl/fl</sup> mice (*Myl1*<sup>Cre</sup>/*Lrrc8a*<sup>fl/fl</sup>), since *Myl1-Cre* is restricted to mature skeletal muscle (*Figure 8—figure supplement 2B*), and excludes brown fat (*Bothe et al., 2000*). Similar to *Myf5*<sup>Cre</sup>/*Lrrc8a*<sup>fl/fl</sup> mice, *Myl1*<sup>Cre</sup>/*Lrrc8a*<sup>fl/fl</sup> mice fed a regular chow diet, have normal glucose tolerance (*Figure 8—figure supplement 2C*), but exhibit 24% reduced exercise capacity on treadmill testing, as compared to WT (*Figure 8—figure*

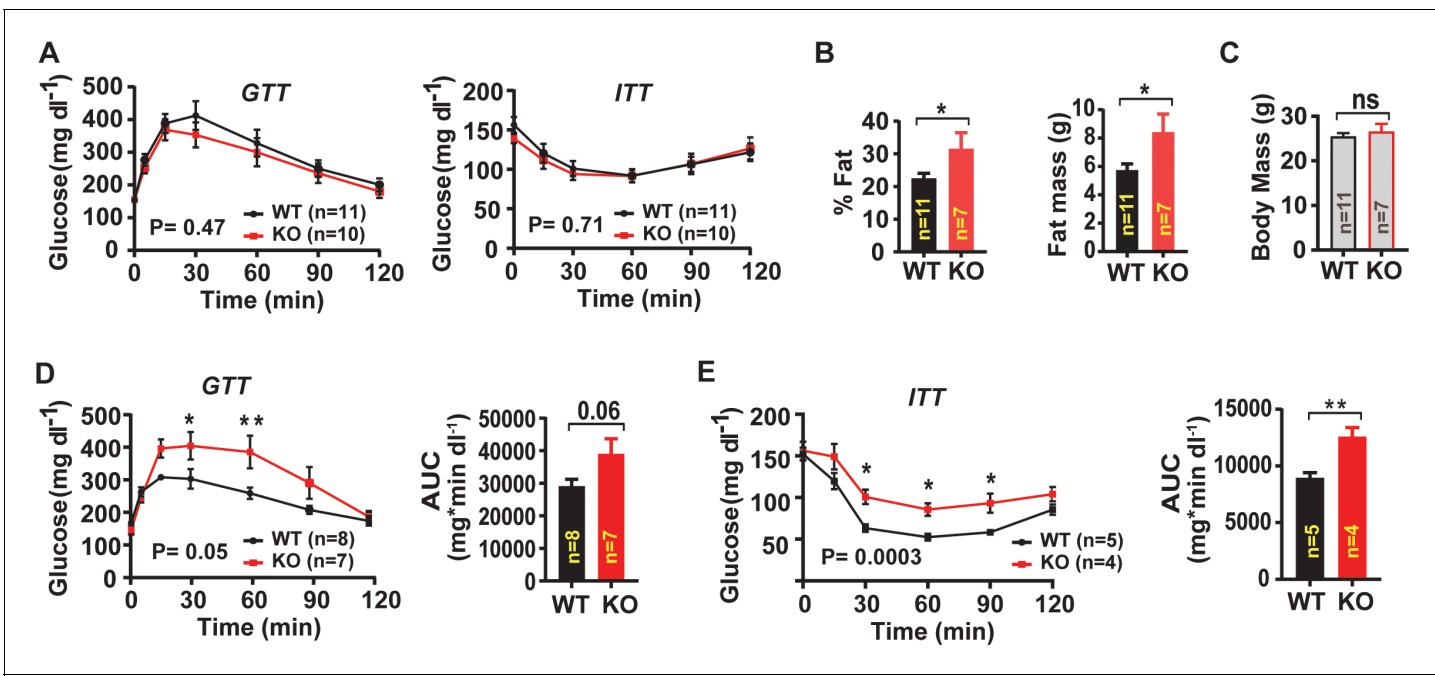

**Figure 8.** Skeletal muscle-targeted *Lrrc8a* ablation increases adiposity and induces glucose intolerance with overnutrition. (A) Glucose and insulin tolerance tests of mice raised on chow diet of WT (n = 11) and KO (n = 10) mice. (B) NMR measurement of fat mass (%) and absolute fat mass of WT (n = 11) and KO (n = 7) mice. (C) Body mass of WT (n = 11) and KO (n = 7) mice on regular chow diet. (D) Glucose tolerance test of WT (n = 8) and KO (n = 7) mice fed HFD for 16 weeks after 14 weeks of age. Corresponding area under the curve (AUC) for glucose tolerance for WT and KO mice. (E) Insulin tolerance tests of WT (n = 5) and KO (n = 4) mice fed HFD for 18 weeks after 14 weeks of age. Corresponding area under the curve (AUC) for insulin tolerance for WT and KO mice. Statistical significance test between the indicated group B, C, D and E (AUC) were calculated by using a two-tailed Student's t-test. Error bars represent mean ± s.e.m. Two-way ANOVA was used for A, D and E (p-value in bottom corner of graph). Error bars represent mean ± s.e.m. *, p<0.05, **, p<0.01, ***, p<0.001.

The online version of this article includes the following figure supplement(s) for figure 8:

**Figure supplement 1.** Body composition of skeletal muscle specific *Lrrc8a* KO mice raised on a high-fat-diet (HFD).

**Figure supplement 2.** Skeletal muscle-targeted *Lrrc8a* ablation impairs muscle endurance and induces adiposity.

supplement 2D). Also, $Myl1^{Cre}/Lrrc8a^{fl/fl}$ mice develop increased visceral adiposity over time on regular chow, based on 24% increased epididymal adipose mass normalized to body mass (*Figure 8—figure supplement 2E*), with no differences in inguinal adipose tissue, muscle mass (*Figure 8—figure supplement 2F*), or total body mass (*Figure 8—figure supplement 2G*). These data suggest that impaired skeletal muscle glucose uptake in Skm KO ($Myl1^{Cre}/Lrrc8a^{fl/fl}$ and $Myf5^{Cre}/Lrrc8a^{fl/fl}$) mice are compensated for by increased adipose glucose uptake and de novo lipogenesis, which contribute to preserved glucose tolerance, at the expense of increased adiposity in skeletal muscle-targeted *Lrrc8a* KO mice raised on a regular chow diet. However, overnutrition-induced obesity, and the associated impairments in adipose and hepatic glucose disposal may uncover glucose intolerance and insulin resistance in skeletal muscle-targeted *Lrrc8a* KO mice.

## Discussion

Our data reveal that the LRRC8A channel complex regulates insulin/stretch-mediated AKT-AS160-GLUT4, MAP kinase and mTOR signaling in differentiated myoblast cultures, with consequent effects on myogenic differentiation, insulin-stimulated glucose metabolism and oxygen consumption. In vivo, skeletal muscle-targeted *Lrrc8a* KO mice have smaller skeletal muscle cells, impaired muscle endurance, and force generation, and are predisposed to adiposity, glucose intolerance and insulin resistance. Insulin/stretch-mediated PI3K-AKT, mTOR signaling are well known to be important regulators of myogenic differentiation (*Rotwein and Wilson, 2009*; *Héron-Milhavet et al., 2008*), metabolism and muscle function (*Schiaffino et al., 2013*) suggesting impaired LRRC8A-AKT-mTOR signaling may underlie the defect in myogenic differentiation. Indeed, consistent with our previous findings and proposed model in adipocytes, in which LRRC8A mediates the interaction of GRB2 with IRS1 to regulate insulin-AKT signaling (*Zhang et al., 2017*; *Gunasekar et al., 2019*), LRRC8A also associates with GRB2 in skeletal myotubes, and GRB2 knock-down rescues impaired myogenic differentiation in *Lrrc8a* KO muscle cells. Thus, our working model for LRRC8A-mediated regulation of insuln-PI3K-AKT and downstream signaling in adipocytes (*Gunasekar et al., 2019*) appears to be conserved in skeletal myotubes. The in vitro phenotype that we observe in CRISPR/cas9-mediated *Lrrc8a KO* C2C12 myotubes and in *Lrrc8a KO* primary myotubes is consistent with the observation of *Chen et al., 2019* that used siRNA-mediated LRRC8A knock-down to demonstrate that the LRRC8A channel complex is required for myogenic differentiation. However, the ability of both GRB2 KD and LRRC8A O/E to rescue myogenic differentiation and augment insulin-AKT, MAP kinase and mTOR signaling in *Lrrc8a* KO myotubes implicates non-canonical, non-conductive signaling mechanisms. Based on our work and also previous studies (*Voss et al., 2014*; *Qiu et al., 2014*), LRRC8A O/E does *not increase* $I_{Cl,SWELL}$/VRAC to supranormal levels at the plasma membrane. However, pAKT, pERK1/2 and mTOR levels are augmented by twofold to threefold *above endogenous* levels, upon twofold LRRC8A O/E in C2C12 myotubes. These data suggest that alternative/non-canonical signaling mechanisms underlie LRRC8A signaling, as opposed to canonical/conductive signaling mechanisms.

Another finding that warrants further study is the requirement for LRRC8A in stretch-induced AKT and MAP kinase signaling in C2C12 myotubes upon static stretching. Mechanical stretch is known to regulate myoblast proliferation and differentiation and myofiber hypertrophy via PI3K-AKT-MAPK signaling (*Schiaffino et al., 2013*; *Ma et al., 2017*; *Fu et al., 2018*). Also, there are numerous reports that VRAC, and presumably LRRC8A , is mechano-responsive (*Browe and Baumgarten, 2003*; *Browe and Baumgarten, 2006*; *Osei-Owusu et al., 2018*; *Barakat et al., 1999*; *Nakao et al., 1999*; *Nilius and Droogmans, 2001*; *Romanenko et al., 2002*; *Strange et al., 2019*). Therefore, it may not be surprising that LRRC8A complexes co-regulate both insulin and stretch-mediated PI3K-AKT, ERK1/2 signaling in skeletal myotubes - potentially integrating mechanical and hormonal stimuli to tune downstream signaling. Indeed, the concept of mechano-tuning insulin signaling has been proposed and demonstrated in other cell systems (*Chen and Chalfie, 2014*; *Kim et al., 2018*) which implicate integrin signaling (*Kim et al., 2018*) as the mechano-sensory mechanism. Curiously, it has been reported that VRAC can be activated in cardiac muscle cells by applying mechanical tension to β1-integrins (*Browe and Baumgarten, 2003*; *Browe and Baumgarten, 2006*), supporting the notion that, in striated muscle, integrin-LRRC8A may indeed participate in mechano-tuning insulin-AKT and downstream signaling. Further studies to more fully delineate the putative molecular mechanisms are necessary.

It is also notable that the profound myogenic differentiation block observed upon *Lrrc8a* ablation in both C2C12 myotubes and primary myotubes in vitro is significantly milder in vivo, where only a 30% reduction in skeletal myocyte cross-sectional area is observed, with no change in total muscle mass, or lean content, in Skm KO mice. This discordance in phenotype may reflect fundamental differences in the biology of skeletal muscle differentiation in vitro versus the in vivo milieu. Alternatively, it may be that, although early, the time interval between *Myf5-Cre* expression early in myogenesis, and ultimate reductions in LRRC8A protein (potentially ~3 days) may extend beyond the critical period during which LRRC8A is required for myogenic differentiation. Directly testing this hypothesis would require examining mice expressing Cre-recombinase at the precursor stage, in skeletal muscle satellite cells, such as Pax3 or Pax7 promoters (*Relaix et al., 2006*; *Buckingham et al., 2006*) - these experiments are currently underway.

Although overall muscle development is grossly intact in both LRRC8A skeletal muscle KO (*Myl1-Cre/Lrrc8a<sup>fl/fl</sup>* and *Myf5<sup>Cre</sup>/Lrrc8a<sup>fl/fl</sup>*) mice, there is a consistent reduction in exercise capacity, muscle endurance and force generation, and a propensity for increased adiposity over time compared to age and gender matched controls. The impaired exercise capacity observed in skeletal muscle *Lrrc8a* KO mice are consistent with some level of insulin resistance, as in *db/db* mice (*Ostler et al., 2014*) and in humans (*Reusch et al., 2013*), and may be due to impaired skeletal muscle glycolysis and oxygen consumption in LRRC8A-depleted skeletal muscle. Furthermore, the increased gonadal adiposity, with preserved glucose and insulin tolerance, observed in LRRC8A skeletal muscle KO (*Myl1<sup>Cre</sup>/Lrrc8a<sup>fl/fl</sup>* and *Myf5<sup>Cre</sup>/Lrrc8a<sup>fl/fl</sup>*) mice phenocopy both skeletal muscle-specific insulin receptor KO mice (MIRKO) (*Brüning et al., 1998*) and transgenic mice expressing a skeletal muscle dominant-negative insulin receptor mutant (*Moller et al., 1996*), wherein skeletal muscle-specific insulin resistance is compensated for by re-distribution of glucose from skeletal muscle to adipose tissue, to promote adiposity (*Kim et al., 2000*). In the case of LRRC8A skeletal muscle KO mice, overnutrition and HFD feeding unmasks this underlying mild insulin resistance and glucose intolerance, as adipose-tissue insulin resistance also begins to set in. Recent findings from skeletal muscle specific AKT1/AKT2 double KO mice indicate that these effects may not attributable to solely to muscle AKT signaling (*Jaiswal et al., 2019*), but potentially involve other insulin-sensitive signaling pathways.

In summary, we show that LRRC8A regulates myogenic differentiation and insulin-PI3K-AKT-AS160, ERK1/2, and mTOR signaling in myotubes via GRB2-mediated signaling. In vivo, LRRC8A is required for maintaining normal exercise capacity, muscle endurance, adiposity under basal conditions, and systemic glycemia in the setting of overnutrition. These findings contribute further to our understanding of LRRC8A channel complexes in the regulation of systemic metabolism.

## Materials and methods

### Key resources table

| Reagent type (species) or resource | Designation | Source or reference | Identifiers | Additional information |
|---|---|---|---|---|
| Genetic reagent (*Mus musculus*) | *Lrrc8a* (Lrrc8a<sup>fl/fl</sup>) | This paper | Sah lab | SWELL1 is a regulator of adipocyte size, insulin signaling and glucose homeostasis (*Zhang et al., 2017*) |
| Strain, strain background (*Mus musculus*) | *Myf5<sup>Cre</sup>* | Jackson lab | *JAX#* 007893, RRID:IMSR_JAX:007893 | |
| Strain, strain background (*Mus musculus*) | *Myl1<sup>Cre</sup>* | Jackson lab | JAX# 24713, RRID:IMSR_JAX:024713 | |
| Cell line (*Mus musculus*) | C2C12 | ATCC | CRL-1772, RRID:CVCL_0188 | |
| Biological sample (*Mus musculus*) | Skeletal muscle primary cell | Lrrc8a<sup>fl/fl</sup> | | Freshly isolated from Lrrc8a<sup>fl/fl</sup> mice |

*Continued on next page*

*Continued*

| Reagent type (species) or resource | Designation | Source or reference | Identifiers | Additional information |
|---|---|---|---|---|
| Antibody | Anti-β-actin (Rabbit monoclonal) | Cell signalling | Cat#8457 s, RRID:AB_10950489 | WB (1:2000) |
| Antibody | Anti-p-AKT1 (Rabbit monoclonal) | Cell signalling | Cat#9018 s, RRID:AB_2629283 | WB (1:1000) |
| Antibody | Anti- Akt1 (Rabbit monoclonal) | Cell signalling | Cat#2938 s, RRID:AB_915788 | WB (1:1000) |
| Antibody | Anti- pAKT2 (Rabbit monoclonal) | Cell signalling | Cat#8599 s, RRID:AB_2630347 | WB (1:1000) |
| Antibody | Anti- Akt2 (Rabbit monoclonal) | Cell signalling | Cat#3063 s, RRID:AB_2225186 | WB (1:1000) |
| Antibody | Anti- pAS160 (Rabbit polyclonal) | Cell signalling | Cat#4288 s, RRID:AB_10545274 | WB (1:1000) |
| Antibody | Anti- AS160 (Rabbit monoclonal) | Cell signalling | Cat#2670 s, RRID:AB_2199375 | WB (1:1000) |
| Antibody | Anti- AMPKα (Rabbit monoclonal) | Cell signalling | Cat#5831 s, RRID:AB_10622186 | WB (1:1000) |
| Antibody | Anti-pAMPKα (Rabbit monoclonal) | Cell signalling | Cat#2535 s, RRID:AB_2106495 | WB (1:1000) |
| Antibody | Anti-FoxO1 (Rabbit monoclonal) | Cell signalling | Cat#2880 s, RRID:AB_2106495 | WB (1:1000) |
| Antibody | Anti-pFoxO1 (Rabbit polyclonal) | Cell signalling | Cat#9464 s, RRID:AB_329842 | WB (1:1000) |
| Antibody | Anti- p70 S6 Kinase (Rabbit polyclonal) | Cell signalling | Cat#9202 s, RRID:AB_331676 | WB (1:1000) |
| Antibody | Anti- p-p70 S6 Kinase (Rabbit polyclonal) | Cell signalling | Cat#9205 s, RRID:AB_330944 | WB (1:1000) |
| Antibody | Anti-pS6 Ribosomal (Rabbit monoclonal) | Cell signalling | Cat#5364 s, RRID:AB_10694233 | WB (1:2000) |
| Antibody | Anti-S6 Ribosomal (Mouse monoclonal) | Cell signalling | Cat#2317 s, RRID:AB_2238583 | WB (1:1000) |
| Antibody | Anti- GAPDH (Rabbit monoclonal) | Cell signalling | Cat#5174 s, RRID:AB_1062202 | WB (1:2000) |
| Antibody | Anti-pErk1/2 (Rabbit polyclonal) | Cell signalling | Cat#9101 s, RRID:AB_331772 | WB (1:1000) |
| Antibody | Anti-Erk1/2 (Rabbit polyclonal) | Cell signalling | Cat#9102 s, RRID:AB_330744 | WB (1:1000) |

*Continued on next page*

*Continued*

| Reagent type (species) or resource | Designation | Source or reference | Identifiers | Additional information |
|---|---|---|---|---|
| Antibody | Anti- Grb2 (Mouse monoclonal) | BD | Cat#610111 s, RRID:AB_397517 | WB (1:1000) |
| Antibody | Anti-flag (Mouse monoclonal) | Sigma-Aldrich | Cat#F3165, RRID:AB_259529 | WB (1:2000) |
| Antibody | Anti-LRRC8A (Rabbit polyclonal) | Pacific antibodies | Custom made | Epitope: QRTKSRIEQGIVDRSE, WB (1:1000), SWELL1 is a glucose sensor regulating β-cell excitability and systemic glycaemia (*Kang et al., 2018*) |
| Antibody | Anti-BA-F8 (Mouse monoclonal) | Developmental Studies Hybridoma Bank, Iowa City | Cat#AB_10572253, RRID:AB_10572253 | IF (1:100) |
| Antibody | Anti-SC-71 (Mouse monoclonal) | Developmental Studies Hybridoma Bank, Iowa City | Cat#AB_2147165, RRID:AB_2147165 | IF (1:100) |
| Antibody | Anti-BF-F3 (Mouse monoclonal) | Developmental Studies Hybridoma Bank, Iowa City | Cat#AB_2266724, RRID:AB_2266724 | IF (1:100) |
| Antibody | Anti-laminin (Rabbit polyclonal) | Abcam | Cat# ab11575, RRID:AB_298179 | IF (1:100) |
| Antibody | Anti-IgG (Normal mouse IgG) | Santa Cruz | Cat# sc-2027, RRID:AB_737197 | WB (1:1000) |
| Antibody | Anti-rabbit-HRP | BioRad | Cat# 170–6515, RRID:AB_11125142 | WB (1:10000) |
| Antibody | Anti-mouse-HRP | BioRad | Cat# 170–5047, RRID:AB_11125753 | WB (1:10000) |
| Recombinant DNA reagent | (Ad5-CMV-mCherry) | University of Iowa viral vector core facility | Ad3518 | |
| Recombinant DNA reagent | Ad5-CMV-Cre-mCherry | University of Iowa viral vector core facility | Ad3494 | |
| Recombinant DNA reagent | Ad5-CAG-LoxP-stop-LoxP-3XFlag-LRRC8A | Vector biolabs | 20180313T#1 | |
| Recombinant DNA reagent | Ad5-U6-m-shGRB2-GFP | Vector biolabs | shADV-260737 | |
| Recombinant DNA reagent | Ad5-U6-shSCR-GFP | Vector biolabs | 1122N | |
| Recombinant DNA reagent | Ad5-CMVmCherry-U6-hLRRC8A-shRNA | Vector biolabs | AD3535 | |
| Recombinant DNA reagent | Ad5-U6-scramble-mCherry | Vector biolabs | 3086 | |
| Commercial assay or kit | RNA isolation (PureLink RNA mini kit) | Invitrogen | 12183018A | |
| Chemical compound, drug | Polybrene (Hexadimethrine bromide) | Sigma Aldrich | H9268 | (4 µg/ml) |

*Continued on next page*

Continued

| Reagent type (species) or resource | Designation | Source or reference | Identifiers | Additional information |
|---|---|---|---|---|
| Software, algorithm | GraphPad Prism8 | | La Jolla California USA, www.graphpad.com' RRID:SCR_002798 | |
| Software, algorithm | Fiji, ImageJ | *Schindelin et al., 2012* (PMID:22743772) | RRID:SCR_002285 | |
| Other | DAPI stain | Invitrogen | D1306 | (1 μg/ml) |

## Animals

The Institutional Animal Care and Use Committee of the Washington University in St. Louis and the University of Iowa approved all experimental procedures involving animals. All the mice were housed in temperature, humidity, and light-controlled room and allowed free access to water and food. Male and female $Lrrc8a^{fl/fl}$ (WT), $Myl1^{Cre}/Lrrc8a^{fl/fl}$, $Myf5^{Cre}/Lrrc8a^{fl/fl}$ (skeletal muscle targeted $Lrrc8a$ KO), were generated and used in these studies. $Myl1Cre$ (JAX# 24713) and $Myf5Cre$ (JAX# 007893) mice were purchased from Jackson labs. For high-fat diet (HFD) studies, we used Research Diets Inc (Cat # D12492) (60 kcal% fat) regimen starting at 14 weeks of age.

## Generation of CRISPR/Cas9-mediated LRRC8A floxed ( $Lrrc8a^{fl/fl}$) mice

$Lrrc8a^{fl/fl}$ mice were generated as previously described (*Zhang et al., 2017*). Briefly, $Lrrc8a$ intronic sequences were obtained from Ensembl Transcript ID ENSMUST00000139454. All CRISPR/Cas9 sites were identified using ZiFit Targeter Version 4.2 (http://zifit.partners.org/ZiFiT/). Pairs of oligonucleotides corresponding to the chosen CRISPR-Cas9 target sites were designed, synthesized, annealed, and cloned into the pX330-U6-Chimeric_BB-CBh-hSpCas9 construct (Addgene plasmid # 42230), following the protocol detailed in *Cong et al., 2013*. CRISPR-Cas9 reagents and ssODNs were injected into the pronuclei of F1 mixed C57/129 mouse strain embryos at an injection solution concentration of 5 ng/μl and 75–100 ng/μl, respectively. Correctly targeted mice were screened by PCR across the predicted loxP insertion sites on either side of Exon 3. These mice were then backcrossed >8 generations into a C57BL/6 background.

## Antibodies

Rabbit polyclonal anti-LRRC8A antibody was generated against the epitope QRTKSRIEQGIVDRSE (Pacific Antibodies). All other primary antibodies were purchased from Cells Signaling: anti-β-actin (#8457 s), p-AKT1 (#9018 s), Akt1 (#2938 s), pAKT2 (#8599 s), Akt2 (#3063 s), p-AS160 (#4288 s), AS160 (#2670 s), AMPKα (#5831 s), pAMPKα (#2535 s), FoxO1(#2880 s) and pFoxO1(#9464 s), p70 S6 Kinase (#9202 s), p-p70 S6 Kinase (#9205 s), pS6 Ribosomal (#5364 s), GAPDH (#5174 s), pErk1/2 (#9101 s), Total Erk1/2 (#9102 s). Purified mouse anti-Grb2 was purchased from BD (610111 s). Purified anti-flag mouse antibody was purchased from sigma. Rabbit IgG Santa Cruz (sc-2027). All primary antibodies were used at 1:1000 dilution, except for anti-flag at 1:2000 dilution. All secondary antibody (anti-rabbit-HRP and anti-mouse-HRP) were used at 1:10,000 dilution.

## Adenovirus

Adenovirus type five with Ad5-CMV-mCherry ($1 \times 10^{10}$ PFU/ml), Ad5-CMV-Cre-mCherry ($3 \times 10^{10}$ PFU/ml) were obtained from the University of Iowa viral vector core facility. Adenovirus type five with Ad5-CMVmCherry-U6-hLRRC8A-shRNA, $2.2 \times 10^{10}$ PFU/ml, (AD3535), (Vector Biolabs). Scrambled non-targeting control (shSCR: Ad5-U6-scramble-mCherry, $1 \times 10^{10}$ PFU/ml), (Vector biolabs #3086). Ad5-CAG-LoxP-stop-LoxP-3XFlag- LRRC8A ($1 \times 10^{10}$ PFU/ml) were obtained from Vector Biolabs. Ad5-U6-shGRB2-GFP ($1 \times 10^{9}$ PFU/ml) and Ad5-U6-shSCR-GFP ($1 \times 10^{10}$ PFU/ml) were obtained from Vector Biolabs.

## Electrophysiology

All recordings were performed in the whole-cell configuration at room temperature, as previously described (*Zhang et al., 2017*; *Kang et al., 2018*). Briefly, currents were measured with either an

Axopatch 200B amplifier or a MultiClamp 700B amplifier (Molecular Devices) paired to a Digidata 1550 digitizer, using pClamp 10.4 software. The intracellular solution contained (in mM): 120 L-aspartic acid, 20 CsCl, 1 MgCl$_2$, 5 EGTA, 10 HEPES, 5 MgATP, 120 CsOH, 0.1 GTP, pH 7.2 with CsOH. The extracellular solution for hypotonic stimulation contained (in mM): 90 NaCl, 2 CsCl, 1 MgCl2, 1 CaCl2, 10 HEPES, five glucose, five mannitol, pH 7.4 with NaOH (210 mOsm/kg). The isotonic extracellular solution contained the same composition as above except for mannitol concentration of 105 (300 mOsm/kg). The osmolarity was checked by a vapor pressure osmometer 5500 (Wescor). Currents were filtered at 10 kHz and sampled at 100 μs interval. The patch pipettes were pulled from borosilicate glass capillary tubes (WPI) using a P-87 micropipette puller (Sutter Instruments). The pipette resistance was ~4–6 MΩ when the patch pipette was filled with intracellular solution. The holding potential was 0 mV. Voltage ramps from −100 to +100 mV (at 0.4 mV/ms) were applied every 4 s.

## Primary muscle satellite cell isolation

Satellite cell isolation and differentiation were performed as described previously with minor modifications (*Hindi et al., 2017*). Briefly, gastrocnemius and quadriceps muscles were excised from *Lrrc8a*$^{flfl}$ mice (8–10 weeks old) and washed twice with 1XPBS supplemented with 1% penicillin-streptomycin and fungizone (300 μl/100 ml). Muscle tissue was incubated in DMEM-F12 media supplemented with collagenase II (2 mg/ml), 1% penicillin-streptomycin and fungizone (300 ul/100 ml) and incubated at shaker for 90 min at 37°C. Tissue was washed with 1X PBS and incubated again with DMEM-F12 media supplemented with collagenase II (1 mg/ml), dispase (0.5 mg/ml), 1% penicillin-streptomycin and fungizone (300 μl/100 ml) in a shaker for 30 min at 37°C. Subsequently, the tissue was minced and passed through a cell strainer (70 μm), and after centrifugation; satellite cells were plated on BD Matrigel-coated dishes. Cells were stimulated to differentiate into myoblasts in DMEM-F12, 20% fetal bovine serum (FBS), 40 ng/ml basic fibroblast growth factor (bfgf, R and D Systems, 233-FB/CF), 1X non-essential amino acids, 0.14 mM β-mercaptoethanol, 1X penicillin/streptomycin, and Fungizone. Myoblasts were maintained with 10 ng/ml bfgf and then differentiated in DMEM-F12, 2% FBS, 1X insulin–transferrin–selenium, when 80% confluency was reached.

## Cell culture

WT C2C12 and *Lrrc8a KO* C2C12 cell line were cultured at 37°C, 5% CO$_2$ Dulbecco's modified Eagle's medium (DMEM; GIBCO) supplemented with 10% fetal bovine serum (FBS; Atlanta Bio selected) and antibiotics 1% penicillin-streptomycin (Gibco, USA). Cells were grown to 80% confluency and then transferred to differentiation media DMEM supplemented with antibiotics and 2% horse serum (HS; GIBCO) to induce differentiation. The differentiation media was changed every two days. Cells were allowed to differentiate into myotubes for up to 6 days. Subsequently, myotube images were taken for quantification of myotube surface area and fusion index.

## Myotube morphology, surface area and fusion index quantification

After differentiation (Day 7), cells were imaged with Olympus IX73 microscope (10X objective, Olympus, Japan). For each experimental condition, 5–6 bright field images were captured randomly from six-well plate. Myotube surface area was quantified manually with ImageJ software. The morphometric quantification was carried out by an independent observer who was blinded to the experimental conditions. For fusion index, differentiated myotube growing on coverslip were washed with 1X PBS and fixed with 2% PFA. After washing with 1XPBS three times, cells were permeabilized with 0.1% TritonX100 for 5 min at room temperature and subsequently blocking was done with 5% goat serum for 30 min. Cells were stained with DAPI (1 μM) for 15 min and after washing with 1X PBS, coverslip were mounted on slides with ProLong Diamond anti-fading agent. Cells were imaged with Olympus IX73 microscope (10X objective, Olympus, Japan) with bright field and DAPI filter. Fusion index (number of nuclei incorporated within the myotube/total number of nuclei present in that view field) were analyzed by ImageJ.

## RNA sequencing

RNA quality was assessed by Agilent BioAnalyzer 2100 by the University of Iowa Institute of Human Genetics, Genomics Division. RNA integrity numbers greater than eight were accepted for RNAseq

library preparation. RNA libraries of 150 bp PolyA-enriched RNA were generated, and sequencing was performed on a HiSeq 4000 genome sequencing platform (Illumina). Sequencing results were uploaded and analyzed with BaseSpace (Illumina). Sequences were trimmed to 125 bp using FASTQ Toolkit (Version 2.2.0) and aligned to *Mus musculus* mmp10 genome using RNA-Seq Alignment (Version 1.1.0). Transcripts were assembled and differential gene expression was determined using Cufflinks Assembly and DE (Version 2.1.0). Ingenuity Pathway Analysis (QIAGEN) was used to analyze significantly regulated genes which were filtered using cutoffs of >1.5 fragments per kilobase per million reads,>1.5 fold changes in gene expression, and a false discovery rate of <0.05. Heatmaps were generated to visualize significantly regulated genes.

## Myotube signaling studies

For insulin stimulation, differentiated C2C12 myotubes were incubated in serum free media for 6 hr and stimulated with 0 and 10 nM insulin for 15 min; while differentiated primary myotubes were incubated in serum free media for 4 hr and stimulated with 0 and 10 nM insulin for 2 hr. To examine intracellular signaling upon LRRC8A overexpression ( LRRC8A O/E), we overexpressed LRRC8A-3xFlag by transducing C2C12 myotubes with Ad5-CAG-LoxP-stop-LoxP-LRRC8A-3xFlag (MOI 50–60) and Ad5-CMV-Cre-mCherry (MOI 50–60) and polybrene (4 μg/ml) in DMEM (2% FBS and 1% penicillin-streptomycin) for 36 hr. Ad5-CMV-Cre-mCherry alone with polybrene (4 μg/ml) (MOI 50–60) was transduced in WT C2C12 or *Lrrc8a* KO C2C12 as controls. Viral transduction efficiency (60–70%) was confirmed by mCherry fluorescence. Cells were allowed to differentiate further in differentiation media up to 6 days. Myotube images were taken before collecting lysates for further signaling studies. GRB2 knock-down was achieved by transducing myotubes with Ad5-U6- shSCR-GFP (Control, MOI 50–60) or Ad5-U6- sh LRRC8A-GFP (GRB2 KD, MOI 50–60) in DMEM (2% FBS and 1% penicillin-streptomycin) supplemented with polybrene (4 μg/ml) for 24 hr. Cells were allowed to differentiate further in differentiation media up to 6 days. Differentiated myotube images were taken for myotube surface area quantification before collecting the cells for RNA isolation.

## Myotube signaling upon LRRC8A disruption post-differentiation

WT C2C12 cells were cultured in differentiation media for 6 days to form myotubes. Subsequently myotubes were transduced either with Ad5-U6-scramble-mCherry (sh-SCR) or Ad5-mCherry-U6-shLRRC8A(sh LRRC8A) shRNA (KD) (MOI 50–60) for 2 days and grown for 1 additional day prior to analysis. Myotubes were then incubated in serum free media for 6 hr and stimulated with 0 and 10 nM insulin for 15 min. Primary skeletal muscle cells isolated from *Lrrc8a* [fl/fl] mice were grown in differentiation media for 3.5 days. To generate *Lrrc8a* KO primary myotubes, cells were transduced with either Ad5-CMV-CMV-mCherry (WT) or Ad5-CMV-Cre-mCherry (KO) (MOI 50–60) for 2 days and then grown for 1 additional day prior to analysis. Myotubes were then harvested to measure intracellular signaling under basal culture conditions.

## Stretch stimulation

C2C12 myotubes were plated in each well of a six-well BioFlex culture plate. Cells were allowed to differentiate up to 6 days in differentiation media, and then placed into a Flexcell Jr. Tension System (FX-6000T) and incubated at 37°C with 5% CO$_2$. C2C12 myotubes on flexible membrane were subjected to either no tension or to static stretch of 5% for 15 min. Cells were lysed and protein isolated for subsequent western blots.

## Western blot

Cells were washed with ice cold 1X PBS and lysed in ice-cold lysis buffer (150 mM NaCl, 20 mM HEPES, 1% NP-40, 5 mM EDTA, pH 7.5) with added proteinase/phosphatase inhibitor (Roche). The cell lysate was further sonicated (20% pulse frequency for 20 s) and centrifuged at 14,000 rpm for 20 min at 4°C. The supernatant was collected and estimated for protein concentration using DC protein assay kit (Bio-Rad). For immunoblotting, an appropriate volume of 4 x Laemmli (Bio-rad) sample loading buffer was added to the sample (10–20 μg of protein), then heated at 90°C for 5 min before loading onto 4–20% gel (Bio-Rad). Proteins were separated using running buffer (Bio-Rad) for 2 hr at 110 V. Proteins were transferred to PVDF membrane (Bio-Rad) and membrane blocked in 5% (w/v) BSA or 5% (w/v) milk in TBST buffer (0.2 M Tris, 1.37 M NaCl, 0.2% Tween-20, pH 7.4) at room

temperature for 1 hr. Blots were incubated with primary antibodies at 4°C overnight, followed by secondary antibody (Bio-Rad, Goat-anti-mouse #170–5047, Goat-anti-rabbit #170–6515, all used at 1:10,000) at room temperature for one hour. Membranes were washed three times and imaged by chemiluminescence (Pierce) by using a Chemidoc imaging system (BioRad). The images were further analyzed for band intensities using ImageJ software. β-Actin or GAPDH levels were quantified for equal protein loading.

## Immunoprecipitation

C2C12 myotubes were plated on 10 cm dishes in complete media and grown to 80% confluency. For LRRC8A-3xFlag overexpression, Ad5-CAG-LoxP-stop-LoxP-3XFlag- LRRC8A (MOI 50–60) and Ad5-CMV-Cre-mCherry (MOI 50–60) along with polybrene (4 ug/ml) were added to cells in DMEM media (2% FBS and 1% penicillin-streptomycin) allowed to grow for 36 hr. Cells were then switched to differentiation media for up to 6 days. After that myotubes were harvested in ice-cold lysis buffer (150 mM NaCL, 20 mM HEPES, 1% NP-40, 5 mM EDTA, pH 7.5) with added protease/phosphatase inhibitor (Roche) and kept on ice with gentle agitation for 15 min to allow complete lysis. Lysated were incubated with anti-Flag antibody (Sigma #F3165) or control rabbit IgG (Santa Cruz sc-2027) rotating end over end overnight at 4°C. Protein G sepharose beads (GE) were added for 4 hr and then samples were centrifuged at 10,000 g for 3 min and washed three times with RIPA buffer and re-suspended in laemmli buffer (Bio-Rad), boiled for 5 min, separated by SDS-PAGE gel followed by the western blot protocol.

## RNA isolation and quantitative RT-PCR

Differentiated cells were solubilized in TRIzol and the total RNA was isolated using PureLink RNA kit (Life Technologies) and column DNase digestion kit (Life Technologies). The cDNA synthesis, qRT-PCR reaction and quantification were carried out as described previously (*Zhang et al., 2017*). All experiments were performed in triplicate and GAPDH were used as internal standard to normalize the data. All primers used for qRT-PCR are listed in *Supplementary file 4*.

## Muscle tissue homogenization

Mice were euthanized and gastrocnemius muscle excised and washed with 1X PBS. Muscles tissue were minced with surgical blade and kept in 8 vol of ice cold homogenization buffer (20 mM Tris, 137 mM NaCl, 2.7 mM KCl, 1 mM $MgCl_2$, 1% Triton X-100, 10% (w/v) glycerol, 1 mM EDTA, 1 mM dithiothreitol, pH 7.8) supplemented with protease/phosphatase inhibitor (Roche). Tissues were homogenized on ice with a Dounce homogenizer (40–50 passes) and incubated for overnight at 4°C with continuous rotation. Tissue lysate was further sonicated in 20 s cycle intervals for 2–3 times and centrifuged at 14,000 rpm for 20 min at 4°C. The supernatant was collected for protein concentration estimation using DC protein assay kit (Bio-Rad). Due to the high content of contractile protein in this preparation, coomassie gel staining was performed to demonstrate equal protein loading, and for quantification normalization of Western blots.

## Tissue histology

Mice were anesthetized with isoflurane followed by cervical dislocation. Tibialis anterior (TA) muscle was carefully excised and gently immersed into the tissue-tek O.C.T medium placed on wooden cork. Orientation of the tissue maintained while embedding in the medium. Subsequently, wooden cork with tissue gently immersed into the liquid N2 pre-chilled isopentane bath for 10–14 s and store at −80°C. Tissue sectioning (10 µm) were done with Leica cryostat and all sections collected on positively charged microscope slide for H and E staining as described earlier (*Bonetto et al., 2015*). Briefly, TA sectioned slides were stained for 2 min in hematoxylin, 1 min in eosin and then dehydrated with ethanol and xylenes. Subsequently, slides were mounted with coverslip and image were taken with EVOS cell imaging microscope (10X objective). For quantification of fiber cross-sectional area, images were processed using ImageJ software to enhance contrast and smooth/sharpen cell boundaries and clearly demarcate muscle fiber cross sectional area. All measurements were performed with an independent observer who was blinded to the identity of the slides.

## Muscle fiber type quantification

Skeletal muscle fiber-type was quantified in WT and Skm KO tibialis anterior (TA) muscle as previously described (*Biltz et al., 2020*). Briefly, sections were immunostained against myosin heavy chain isoforms (type I, type IIa, and type IIb BA-F8, SC-71 and BF-F3, respectively, Developmental Studies Hybridoma Bank, Iowa City, IA) and against laminin (ab11575; Abcam, Cambridge, UK) to quantify fiber types and areas. Fiber types were identified on four non-overlapping 20 × images in controlled locations: two from the superficial region and two from the deep. Unstained fibers were assumed to be of type IIx. Fiber areas were detected with a custom ImageJ macro using the automated Huang threshold and particle analyzer. Regions of interest with a circularity of less than 0.5 were excluded as being out of the axial plane of that fiber.

## Exercise tolerance test and inversion testing

Mouse treadmill exercise protocols were adapted from *Dougherty et al., 2016*. Briefly, mice were first acclimated with the motorized treadmill (Columbus Instruments Exer3/6 Treadmill Columbus, OH) for 3 days by running 10–15 min (with 3 min interval) for three consecutive days at 7 m/min, with the electric shocking grid (frequency 1 Hz) installed in each lane. During the treadmill testing, mice ran with a gradual increase in speed (5.5 m/minute to 22 m/minute) and inclination (0°-15°) at time intervals of 3 min each. The total running distance for each mouse was recorded at the end of the experiment. The predefined criteria for removing the mouse from the treadmill and recording the distance travelled was: continuous shock for 5 s or receiving 5–6 shocks within a time interval of 15 s. These mice were promptly removed from the treadmill and total duration and distance were recorded for further analysis.

Mouse inversion test was performed using a wire-grid screen apparatus elevated to 50 cm. Mice were stabilized on the screen inclined at 60°, with the mouse head facing towards the base of the screen. The screen was slowly pivoted to 0° (horizontal), such that the mouse was fully inverted and hanging upside down from the screen. Soft bedding was placed underneath the screen to protect mouse from any injury, were they to fall. The inversion test for each mouse was repeated two times with an interval of 45 min (resting period). The hang time for each mouse was repeated three times with an interval of 5 min. The maximum hanging time limit for each mouse was set for 3 min.

## Isolated muscle contractile assessment

Soleus muscle was carefully dissected and transferred to a specialized muscle stimulation system (1500A, Aurora Scientific, Aurora, ON, Canada) where physiology tests were run in a blinded fashion. Muscle was immersed in a Ringer solution (in mM) (NaCl 137, KCl 5, CaCl$_2$ 2, NaH$_2$PO$_4$ 1, NaHCO$_3$ 24, MgSO$_4$ 1, glucose 11 and curare 0.015) maintained at 37°C. The distal tendon was secured with silk suture to the arm of a dual mode ergometer (300C-LR, Aurora Scientific, Aurora, ON, Canada) and the proximal tendon secured to a stationary post. Muscles were stimulated with an electrical stimulator (701C, Aurora Scientific, Aurora, ON, Canada) using parallel platinum plate electrodes extending along the muscle. Muscle slack length was set by increasing muscle length until passive force was detectable above the noise of the transducer and fiber length was measured through a micrometer reticule in the eyepiece of a dissecting microscope. Optimal muscle length was then determined by incrementally increasing the length of the muscle by 10% of slack fiber length until the isometric tetanic force plateaued. At this optimum length, force was recorded during a twitch contraction and isometric tetanic contraction (300 ms train of 0.3 ms pulses at 225 Hz). The muscle was then fatigued with a bout of repeated tetanic contractions every 10 s until force dropped below 50% of peak. At this point, the muscle was cut from the sutures and weighed. This weight, along with peak fiber length and muscle density (1.056 g/cm3), was used to calculate the physiological cross-sectional area (PCSA) and convert to specific force (tension). The experimental data were analyzed and quantified using Matlab (Mathworks), and presented as peak tetanic tension (Tetanic Tension) – peak of the force recording during the tetanic contraction, normalized to PCSA; Time to fatigue (TTF) – time for the tetanic tension to fall below 50% of the peak value during the fatigue test; Half relaxation time (HRT) – half the time between force peak and return to baseline during the twitch contraction.

## XF-24 seahorse assay

Cellular respiration was quantified in primary myotubes using the XF24 extracellular flux (XF) bioanalyzer (Agilent Technologies/Seahorse Bioscience, North Billerica, MA, USA). Primary skeletal muscle cells isolated from *Lrrc8a* $^{fl/fl}$ mice were plated on BD Matrigel-coated plate at a density of $20 \times 10^3$ per well. After 24 hr, cells were incubated in Ad5-CMV-mCherry or Ad5-CMV-Cre-mCherry (MOI 90–100) in DMEM-F12 media (2% FBS and 1% penicillin-streptomycin) for 24 hr. Cells were then switched to differentiation media for another 3 days. For insulin-stimulation, cells were incubated in serum free media for 4 hr and stimulated with 0 and 10 nM insulin for 2 hr. Subsequently, medium was changed to XF-DMEM, and kept in a non-CO$_2$ incubator for 60 min. The basal oxygen consumption rate (OCR) was measured in XF-DMEM. Subsequently, oxygen consumption was measured after addition of each of the following compounds: oligomycin (1 µg/ml) (ATP-Linked OCR), carbonyl cyanide 4-(trifluoromethoxy) phenylhydrazone (FCCP; 1 µM) (Maximal Capacity OCR) and antimycin A (10 µM; Spare Capacity OCR) (*Wende et al., 2015*). For the glycolysis stress test, prior to experimentation, cells were switched to glucose-free XF-DMEM and kept in a non-CO2 incubator for 60 min. Extracellular acidification rate (ECAR) was determined in XF-DMEM followed by these additional conditions: glucose (10 mM), oligomycin (1 µM), and 2-DG (100 mM). Data for Seahorse experiments (normalized to protein) reflect the results of one Seahorse run/condition with six replicates.

## Metabolic phenotyping

Mouse body composition (fat and lean mass) was measured by nuclear magnetic resonance (NMR; Echo-MRI 3-in-1 analyzer, EchoMRI, LLC). For glucose tolerance test (GTT), mice were fasted for 6 hr and intraperitoneal injection of glucose (1 g/kg body weight for lean mice and 0.75 g/kg of body weight for HFD mice) administered. Glucose level was monitored from tail-tip blood using a glucometer (Bayer Healthcare LLC) at the indicated times. For insulin tolerance test (ITT), mice were fasted for 4 hr and after an intra-peritoneal injection of insulin (HumulinR, 1 U/kg for lean mice and 1.25 U/kg for HFD mice) glucose level was measured by glucometer at the indicated times.

## Statistics

Data are represented as mean ± s.e.m. Two-tail paired or unpaired Student's t-tests were used for comparison between two groups. For three or more groups, data were analyzed by one-way analysis of variance and Tukey's post hoc test. For GTTs and ITTs, two-way analysis of variance (Anova) was used. A p-value<0.05 was considered statistically significant. *, ** and *** represents a p-value less than 0.05, 0.01 and 0.001 respectively.

## Acknowledgements

RNA-Seq data presented herein were obtained at the Genomics Division of the Iowa Institute of Human Genetics. This work was supported by grants NIH/NHLBI R01 HL125436 (CEG), NIH RO1HL127764, RO1HL112413 (EDA), NIH NIDDK 1R01DK106009 (RS), the Roy J Carver Trust (RS) and Musculoskeletal Research Center Pilot Grant 1P30AR074992-01 (RS).

## Additional information

### Funding

| Funder | Grant reference number | Author |
| --- | --- | --- |
| National Institutes of Health | 1R01DK106009 | Rajan Sah |
| National Institutes of Health | R01HL125436 | Chad E Grueter |
| National Institutes of Health | RO1HL127764 | E Dale Abel |
| National Institutes of Health | RO1HL112413 | E Dale Abel |
| National Institutes of Health | 1P30AR074992-01 | Rajan Sah |
| Roy J. Carver Charitable Trust | | Rajan Sah |

The funders had no role in study design, data collection and interpretation, or the decision to submit the work for publication.

## Author contributions

Ashutosh Kumar, Data curation, Formal analysis, Validation, Investigation, Visualization, Methodology, Writing - original draft, Writing - review and editing; Litao Xie, Karen Shen, Formal analysis, Investigation, Methodology; Chau My Ta, Antentor O Hinton, Formal analysis, Investigation, Visualization, Methodology; Susheel K Gunasekar, Rachel A Minerath, Investigation, Visualization, Methodology; Joshua M Maurer, Formal analysis, Methodology; Chad E Grueter, Formal analysis, Supervision, Visualization, Methodology, Writing - review and editing; E Dale Abel, Supervision, Funding acquisition, Writing - review and editing; Gretchen Meyer, Formal analysis, Supervision, Funding acquisition, Visualization, Writing - review and editing; Rajan Sah, Conceptualization, Resources, Formal analysis, Supervision, Visualization, Writing - original draft, Project administration, Writing - review and editing

## Author ORCIDs

E Dale Abel (iD) https://orcid.org/0000-0001-5290-0738
Rajan Sah (iD) https://orcid.org/0000-0003-1092-1244

## Ethics

Animal experimentation: This study was performed in accordance with the recommendations in the Guide for the Care and Use of Laboratory Animals of the National Institutes of Health. All of the animals were handled according to the approved institutional animal care and use committee (IACUC) protocols of Washington University in St. Louis (20180217) and the University of Iowa (1308148).

## Decision letter and Author response

Decision letter https://doi.org/10.7554/eLife.58941.sa1
Author response https://doi.org/10.7554/eLife.58941.sa2

# Additional files

## Supplementary files

- Supplementary file 1. RNA sequencing data.
- Supplementary file 2. IPA canonical pathway analysis.
- Supplementary file 3. Genotypes from *Myf5-Cre x SWELL1*flfl*breeding*.
- Supplementary file 4. Primers used for qRT-PCR of muscle differentiation gene.
- Transparent reporting form

## Data availability

We have uploaded the RNA sequencing data file in GEO (accession code: Series GSE156667).

The following dataset was generated:

| Author(s) | Year | Dataset title | Dataset URL | Database and Identifier |
|---|---|---|---|---|
| Sah R, Grueter CE | 2020 | Effect of SWELL1 (LRRC8a) gene deletion on gene expression in differentiated myotubes | https://www.ncbi.nlm.nih.gov/geo/query/acc.cgi?acc=GSE156667 | NCBI Gene Expression Omnibus, GSE156667 |

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
