## [Decision Letter]

Thank you for submitting your article "SWELL1-LRRC8 complex regulates skeletal muscle cell size, intracellular signalling, adiposity and glucose metabolism" for consideration by *eLife*. Your article has been reviewed by three peer reviewers, including Muthu Periasamy as the Reviewing Editor and Reviewer #1, and the evaluation has been overseen by Didier Stainier as the Senior Editor.

The reviewers have discussed the reviews with one another and the Reviewing Editor has drafted this decision to help you prepare a revised submission.

Summary:

The manuscript by Kumar et al. assesses the role of SWELL1 in regulating skeletal muscle differentiation, growth and metabolism. By knocking-out and rescuing SWELL1 expression in muscle cells, the authors show that this protein is crucial for myogenesis and myotube formation, and it plays a role in regulating insulin-dependent pathways and stretch-induced signaling. Mechanistically, the authors postulate SWELL1 interacts with and restrains GRB2 to regulate downstream AKT/MAPK pathways. Finally, a muscle-specific KO of SWELL1 exhibits smaller myofibers and reduced muscle endurance with no effect over the total muscle mass. The authors also show that SWELL1 regulates insulin signaling and systemic glucose homeostasis during metabolic challenge with a high fat diet. Overall, the study is well done and the majority of the data is convincing. The study provides interesting insights into potential role of stretch activated channel in skeletal muscle function. While the study is interesting, there are limitations in the manuscript, particularly the link between insulin signaling, and metabolic homeostasis is not thoroughly investigated. Several questions remain to be clarified and some additional data would enhance the study for the readers of *eLife*.

Essential Revisions:

1) Figure 1D- RNA-Seq results could be secondary to differentiation rather than to say that SWELL1 regulates these pathways to influence differentiation. Can the authors comment on this.

2) Figure 2- This is related to the comment in Figure 1D. Are the partial responses to insulin due to the difference in differentiated vs. undifferentiated myocytes? Another question is in the differentiated SWELL1 KO myocytes, what is the impact of SWELL1 absence? Do they fully respond to insulin? So once differentiated, is SWELL1 dispensable for normal function. The pAS160 data suggest there is still a defect whereas the pAKT2 data may signal otherwise. In addition, there certainly may be SWELL1-dependent and SWELL1-independent pathways.

3a) Are the effects of SWELL1 manipulation on insulin signaling or RNA-seq linked to impaired differentiation or the other way around? This question remains outstanding. Perhaps can be included in the Discussion. b) The in vivo effects are not associated with muscle size, but could be due to other effects of SWELL1 KO, such as fiber type switch, impaired oxidative metabolism, impaired insulin signaling.

4) How do the authors explain impaired performance? Can the authors measure some key markers quickly to resolve this issue? NADH staining, fiber typing, gene expression, westerns?

---

## [Author Response]

Essential Revisions:

We have addressed all the major comments (see below) and feel this has strengthened the manuscript and clarified our conclusions. The primary Major comments can be grouped into 2 main questions: 1. Are the marked defects in signaling observed in SWELL1 KO myotubes compared to WT simply due to impaired myotube differentiation in SWELL1 KO myotubes, and associated impaired signaling, or is SWELL1 required for signaling in differentiated myotubes? 2. Is the impaired exercise and muscle performance related to fiber-type switching, or other mechanisms.

1) Are the marked defects in signaling observed in SWELL1 KO myotubes compared to WT simply due to impaired myotube differentiation in SWELL1 KO myotubes, and associated impaired signaling, or is SWELL1 required for signaling in differentiated myotubes? The Major points below fall under this category:

1) Figure 1D- RNA-Seq results could be secondary to differentiation rather than to say that SWELL1 regulates these pathways to influence differentiation. Can the authors comment on this.2) Figure 2- This is related to the comment in Figure 1D. Are the partial responses to insulin due to the difference in differentiated vs. undifferentiated myocytes? Another question is in the differentiated SWELL1 KO myocytes, what is the impact of SWELL1 absence? Do they fully respond to insulin? So once differentiated, is SWELL1 dispensable for normal function. The pAS160 data suggest there is still a defect whereas the pAKT2 data may signal otherwise. In addition, there certainly may be SWELL1-dependent and SWELL1-independent pathways.3a) Are the effects of SWELL1 manipulation on insulin signaling or RNA-seq linked to impaired differentiation or the other way around? This question remains outstanding. Perhaps can be include in the Discussion.

We addressed these questions in 2 ways: 1) We first fully differentiated C2C12 myoblasts into myotubes and then used adenoviral SWELL1 shRNA knock-down (KD) to examine the effect of SWELL1 depletion on insulin-stimulated signaling in differentiated C2C12 myotubes (Figure 2—figure supplement 1A-C). 2) We fully differentiated SWELL1^fl/fl^ primary myotubes in culture and then used Adenoviral-Cre mediated SWELL1 recombination to ablate SWELL1 in differentiated myotubes and then examined basal levels of the same signaling pathways (Figure 2—figure supplement 1D-F). As shown in Figure 2—figure supplement 1, and now described in the Results subsection “LRRC8A regulates multiple insulin dependent signaling pathways in skeletal myotubes”, we observe similar reductions in insulin-stimulated signaling in differentiated C2C12 myotubes, but these differences are, in some cases, less pronounced than observed when SWELL1 is deleted prior to differentiation. These findings suggest that *both* SWELL1-dependent impairments in myotube differentiation *and* direct contributions of SWELL1 to intracellular signaling contribute to the observed signaling defects. Since, in the current version of the manuscript, we describe this in the aforementioned Results subsection with additional experimental evidence (Figure 2—figure supplement 1), we have not included further discussion in the Discussion section.

The observation by the reviewers that pAS160 is quite significantly suppressed upon SWELL1 deletion, while pAKT2 is only partially suppressed is astute. Moreover, this appears to be true not only in SWELL1 depleted skeletal myoblasts that are subsequently differentiated into myotubes (Figure 2), but also in fully differentiated myotubes that are subjected to SWELL1 depletion (Figure 2—figure supplement 1). As the reviewer suggests, this finding is consistent with AKT-independent mechanisms of AS-160 phosphorylation, such as via PKC (Farese, R.V., Am J Physiol: Endocrinology and Metabolism 2002).

2) Is the impaired exercise and muscle performance related to fiber-type switching, or other mechanisms. The Major points below fall under this category:

b) The in vivo effects are not associated with muscle size, but could be due to other effects of SWELL1 KO, such as fiber type switch, impaired oxidative metabolism, impaired insulin signaling.4) How do the authors explain impaired performance? Can the authors measure some key markers quickly to resolve this issue? NADH staining, fiber typing, gene expression, westerns?

To address this question, we stained frozen sections that we had previously stored from both superficial and deep tibialis anterior muscle from skeletal-muscle SWELL1 KO and WT mice for muscle fiber-types (MHC Type 1, MHC Type IIa, MHC Type IIx and MHC Type IIb) in both superficial and deep tibialis anterior muscle in Skm KO as compared to WT mice as described in the Results section subsection “Skeletal muscle targeted *Lrrc8a* knock-out mice have reduced skeletal myocyte size, muscle endurance and ex vivo force generation” and shown in Figure 7—figure supplement-1. We observe no differences in muscle fiber-type between WT and Skm KO mice indicating that observed reductions in muscle force and endurance is not due to muscle fiber-type switching. Indeed, it is increasingly appreciated that changes in muscle metabolic potential and morphology may occur independent from adaptive fiber-type switching as measured by a change in MHC expression (Egan and Zierath, 2013). Instead, we believe that alternative mechanisms are responsible for these observations in SWELL1 KO muscle, such as impaired glycolysis and glucose metabolism (as described in the aforementioned subsection, and Figure 7H), and this is the currently the subject of ongoing studies.